# Promoting anti-tumor immunity by targeting TMUB1 to modulate PD-L1 polyubiquitination and glycosylation

Chengyu Shi [1,2,3,21], Ying Wang[1,2,3,21], Minjie Wu[1,2,3,21], Yu Chen[1,2,3], Fangzhou Liu[1,2,3], Zheyuan Shen[4,5], Yiran Wang[1], Shaofang Xie[6], Yingying Shen[7], Lingjie Sang [1], Zhen Zhang[1], Zerui Gao[1], Luojia Yang[1], Lei Qu[1], Zuozhen Yang[1], Xinyu He[1], Yu Guo[5], Chenghao Pan[4,5], Jinxin Che[5], Huaiqiang Ju [8], Jian Liu [9], Zhijian Cai [7], Qingfeng Yan [1], Luyang Yu [1], Liangjing Wang [10], Xiaowu Dong[4,5], Pinglong Xu [11], Jianzhong Shao [1], Yang Liu[7], Xu Li [6]✉, Wenqi Wang [12]✉, Ruhong Zhou [1,2,13,14,15]✉, Tianhua Zhou [2,16,17]✉ & Aifu Lin [1,2,3,18,19,20]✉

Immune checkpoint blockade therapies targeting the PD-L1/PD-1 axis have demonstrated clear clinical benefits. Improved understanding of the underlying regulatory mechanisms might contribute new insights into immunotherapy. Here, we identify transmembrane and ubiquitin-like domain-containing protein 1 (TMUB1) as a modulator of PD-L1 post-translational modifications in tumor cells. Mechanistically, TMUB1 competes with HECT, UBA and WWE domain-containing protein 1 (HUWE1), a E3 ubiquitin ligase, to interact with PD-L1 and inhibit its polyubiquitination at K281 in the endoplasmic reticulum. Moreover, TMUB1 enhances PD-L1 N-glycosylation and stability by recruiting STT3A, thereby promoting PD-L1 maturation and tumor immune evasion. TMUB1 protein levels correlate with PD-L1 expression in human tumor tissue, with high expression being associated with poor patient survival rates. A synthetic peptide engineered to compete with TMUB1 significantly promotes antitumor immunity and suppresses tumor growth in mice. These findings identify TMUB1 as a promising immunotherapeutic target.

Tumor cells can circumvent immune recognition through immune editing[1]. Immune editing involves the modulation of immune checkpoint molecules, such as programmed cell death ligand-1 (PD-L1)[2]. The interaction between PD-1, expressed on T cells, and PD-L1 on tumor cells inhibits the activation and expansion of CD8+ T cells, thereby enabling cancer cells to evade immune destruction[3,4]. Immune checkpoint blockade therapies targeting the PD-L1/PD-1 axis have demonstrated clinical benefits[5,6]. However, most patients exhibit poor responses to these therapies, which is often attributable to the immunosuppressive nature of the tumor microenvironment[7,8].

Therefore, it is of great clinical and scientific significance to investigate the molecular mechanisms underlying the regulation of PD-L1 and to develop strategies to enhance the efficacy of anti-PD-1/PD-L1 therapies.

The expression of PD-L1 on tumor cells is regulated by various factors, including several post-translational modifications, such as glycosylation, ubiquitination, palmitoylation, and phosphorylation[9–12]. Ubiquitination is crucial for several physiological processes, including cell viability and differentiation and innate and adaptive immunity[13]. Ubiquitination and deubiquitination of PD-L1 regulate its proteasomal degradation, which, in turn, affects PD-1/PD-L1-mediated

A full list of affiliations appears at the end of the paper. ✉e-mail: lixu@westlake.edu.cn; wenqiw6@uci.edu; rhzhou@zju.edu.cn; tzhou@zju.edu.cn; linaifu@zju.edu.cn

immunosuppression[14]. Glycosylation occurs inside the endoplasmic reticulum (ER) and Golgi apparatus in a well-regulated stepwise manner and influences the folding, stability, transport, and function of proteins[15]. The N-glycosylation of PD-L1 is important for the resistance of this protein to ubiquitination-mediated proteasomal degradation, as well as anti-PD-L1 antibody-binding affinity and signaling intensity[16]. Incorrectly glycosylated and misfolded proteins, including PD-L1, are recognized by the ER-associated protein degradation (ERAD) complex, polyubiquitinated by the E3 ligase, and finally, translocated to the cytoplasm for degradation, thereby affecting the function of PD-L1 as an immune checkpoint molecule[17,18]. Targeting the post-translational modifications of PD-L1 may prove to be effective approach for anti-tumor therapy. Further elucidating effector molecules associated with the post-translational modifications of PD-L1 may shed light on the discovery of immunotherapeutic targets for cancer treatment.

Transmembrane and ubiquitin-like domain-containing protein 1 (TMUB1), also referred to as hepatocyte odd protein shuttling (HOPS), is a transmembrane ubiquitin-like protein that was originally identified in the regenerated liver[19]. Accumulating evidence suggests that, in addition to being involved in cell proliferation, TMUB1 is a critical regulator of apoptosis, genomic stability, and cancer development[20]. TMUB1 has been reported to regulate the expression of several important proteins, such as p53, p19$^{Arf}$ and NF-κB[21–23]. As an ER membrane protein, TMUB1 is recognized as a member of the RNF185/Memralin ubiquitination complex and involved in the degradation of different ER-resident proteins, such as HMG-CoA reductase, demonstrating a strong association with the ERAD pathway[24,25].

In the present study, proteins upregulated in breast cancer and associated with CD8$^+$ T cell infiltration were screened for the capability of interacting with PD-L1 using co-immunoprecipitation coupled with mass spectrometry. Among them, TMUB1 is revealed as a positive regulator of PD-L1 by inducing its protein stability. In the ER, TMUB1 binds to PD-L1 to prevent the interaction of the latter with the E3 ligase HECT, UBA and WWE domain-containing protein 1 (HUWE1). Moreover, TMUB1 promotes PD-L1 glycosylation by recruiting the catalytic subunit of the oligosaccharyltransferase (OST) complex STT3A, thus enabling PD-L1 in escaping ERAD. TMUB1 deficiency increases cancer cell sensitivity to T cell-mediated cytotoxicity and enhances antitumor immune response in mice. Moreover, we show that a peptide targeting the interaction between TMUB1 and PD-L1 can promote anti-tumor immune responses in mouse models. Collectively, our study demonstrates that TMUB1 regulates the cellular abundance of PD-L1 to promote cancer cell evasion and may represent a potential target for immunotherapy.

## Results

### Identification of TMUB1 as a positive regulator of PD-L1

We initially carried out Gene Set Enrichment Analysis (GSEA) of PD-L1 based on expression data of invasive breast cancer (BRCA) from The Cancer Genome Atlas (TCGA) and found that some biological process pathways associated with post-translational modifications were enriched (Fig. 1a). Recent studies have revealed that PD-L1 in tumor cells undergoes various post-translational modifications such as phosphorylation, acetylation, palmitoylation, glycosylation, and ubiquitination, which regulate PD-L1 quality control, protein stability, and nuclear translocation, ultimately affecting the immune escape of tumor cells[9,10,18,26–29]. Based on the mass spectrometry datasets of tumors and non-malignant tissues from breast cancer patients, we found higher levels of PD-L1 glycosylation and phosphorylation in tumors than that in non-malignant tissues[30,31] (Fig. 1b, c). Furthermore, by comparing the pathway of anti-PD-L1 treatment non-responders with baseline in the 4T1 breast cancer mouse model from TISMO (http://tismo.cistrome.org/)[32] (Supplementary Fig. 1a–e), we found that several of these post-translational modification pathways were upregulated, implying that post-translational modifications of PD-L1 might be a feature of tumors and promising target for immunotherapy.

To identify potential effector molecules associated with PD-L1 post-translational modifications, HEK-293T cell lines with a stable PD-L1-Flag expression were established. Using immunoprecipitation coupled with mass spectrometry, we identified several proteins capable of interacting with PD-L1 (Supplementary Table 1). After comparing these potential PD-L1-interacting molecules with genes upregulated and negatively associated with tumor CD8$^+$ T cell infiltration[33] in the TCGA BRCA dataset (Supplementary Table 2), TMUB1 was found to be an interesting candidate for further investigation (Fig. 1d and Supplementary Fig. 1f, g). Immunohistochemistry (IHC) and RT-qPCR analysis of the tissue samples from a cohort of individuals with breast cancer obtained from Sun Yat-sen University Cancer Center (Supplementary Table 3) showed that high levels of TMUB1 and PD-L1 were significantly associated with decreased survival, and the CD8$^+$ T cell infiltration in the patients' tumor tissue was negatively correlated with the protein level of TMUB1 (Fig. 1e–g and Supplementary Fig. 1h). These findings suggest that TMUB1 could regulate PD-L1 and facilitate the immune escape of tumors.

The interaction between TMUB1 and endogenous PD-L1 in MDA-MB-231 cells was verified using the immunoprecipitation assay, which was further confirmed using the ectopic expressions of TMUB1 and PD-L1 in MDA-MB-231 cells and HEK-293T cells (Fig. 1h and Supplementary Fig. 1g, l). Additionally, the His-tagged TMUB1 and GST-tagged PD-L1 were isolated from *E. coli* and subjected to the in vitro pull-down assay, which showed a direct binding between these two proteins (Supplementary Fig. 1i, j). Recent studies have demonstrated that TMUB1 could modulate the stability of the E3 ligase TRAF6 through direct binding[21]. Similarly, TMUB1 acted as a stable regulator of PD-L1 in MDA-MB-231 cells. Moreover, overexpression of TMUB1 led to an increase of PD-L1 (Supplementary Fig. 1l) and this regulation of PD-L1 by TMUB1 was in a dose-dependent manner (Supplementary Fig. 1k). On the other hand, TMUB1 knockdown reduced the PD-L1 expression (Supplementary Fig. 1m). Similar results were observed in the A549 non-small cell lung cancer cell line, the HepG2 liver cancer cell line, and the AGS gastric cancer cell line (Supplementary Fig. 1n–r)

Membrane-localized PD-L1 inhibits the antitumor immunity by binding to PD-1 located on T cells[34]. Therefore, we examined whether TMUB1 regulates the level of PD-L1 on the plasma membrane. Immunofluorescence, flow cytometry and membrane separation, revealed that overexpression of TMUB1 significantly increased the amount of the PD-L1 protein on the cell membrane (Fig. 1i and Supplementary Fig. 2a, b). Conversely, TMUB1 knockdown led to a decrease of membrane-localized PD-L1 (Fig. 1j).

Considering that the expression of PD-L1 can either be constitutive expression or induced, we further investigated the association between TMUB1 and other PD-L1 regulators, such as IFN-γ and Myc[35,36]. Changes in TMUB1 levels in breast cancer cells did not affect the RNA levels of PD-L1 (Supplementary Fig. 2c and 2d). The mRNA level of TMUB1 did not change significantly under the stimulation of IFN-γ and overexpression of c-Myc (Supplementary Fig. 2e and 2f), respectively, and the interaction between TMUB1 and PD-L1 was not affected, suggesting that these PD-L1 regulators do not regulate TMUB1, and that TMUB1 is an independent factor for PD-L1 regulation (Supplementary Fig. 2g). In addition, there is no cross-talk between TMUB1 and CMTM4/CMTM6 in the regulation of PD-L1 (Supplementary Fig. 2i–q). On the other hand, TMUB1 similarly regulates the cytokine-mediated expression of PD-L1. The knockdown of TMUB1 significantly antagonizes the increase in total PD-L1 abundance and PD-L1 membrane levels caused by IFN-γ stimulation in the MDA-MB-231 cell line (Fig. 1k and Supplementary Fig. 2h).

To investigate whether this regulatory effect of TMUB1 on PD-L1 influences the immune escape of tumor cells, a T cell killing assay was performed using wild-type and stable TMUB1-knockdown MDA-MB-231 and MDA-MB-468 breast cancer cell lines co-cultured with

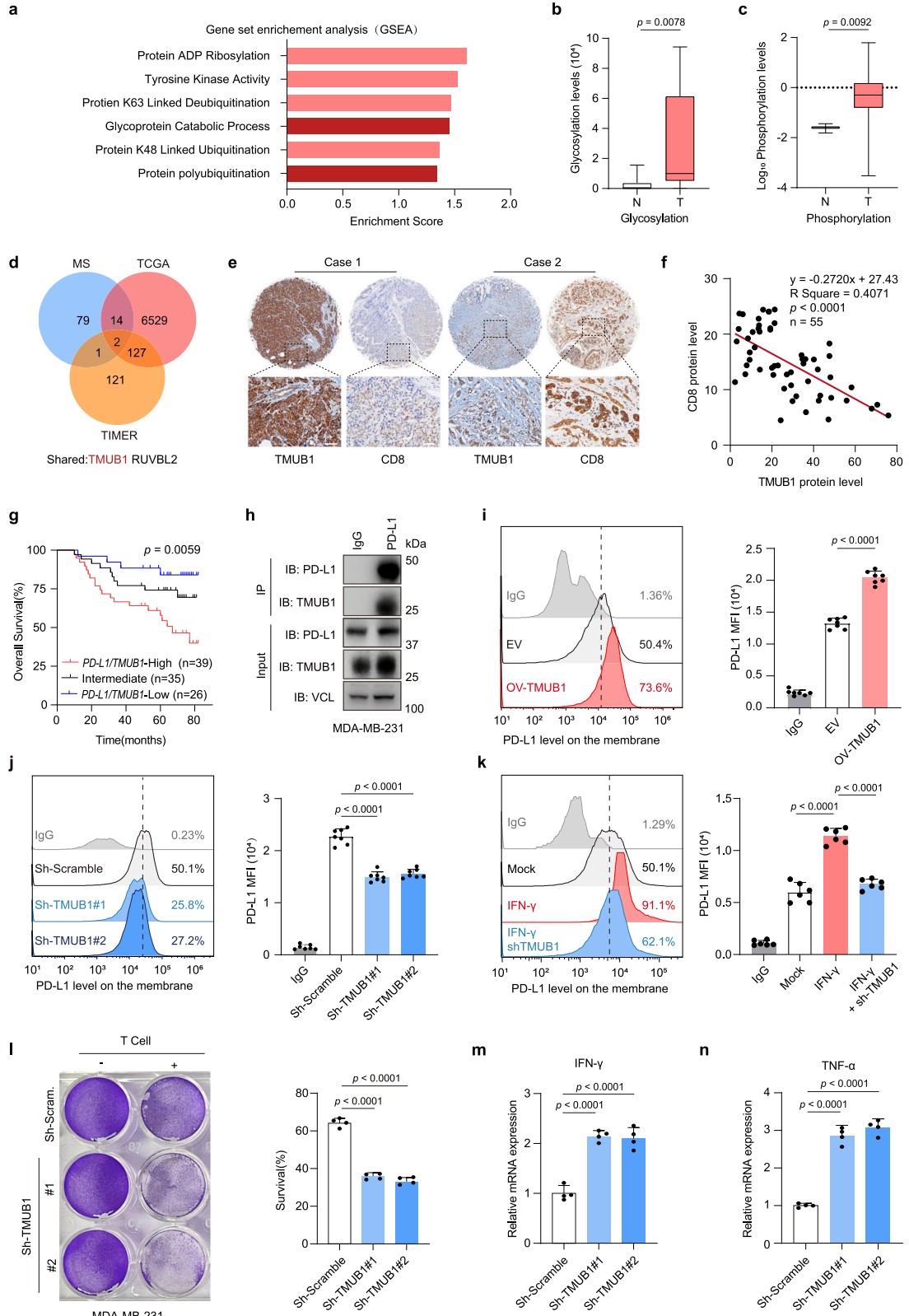

activated peripheral blood mononuclear cells (PBMCs). Interestingly, TMUB1 knockdown attenuated the tumor cell-induced immune suppression, leading to an increased T cell-mediated death of cancer cells (Fig. 1l and Supplementary Fig. 2r). In addition, the expressions of IFN-γ and TNF-α were both increased in the PBMCs co-cultured with the TMUB1 downregulated MDA-MB-231 cells compared to those co-cultured with the wild-type cells (Fig. 1m, n and Supplementary Fig. 2s,

t). Collectively, these results suggested that TMUB1 could increase the PD-L1 levels in cancer cells and promote immune evasion.

## TMUB1 stabilizes PD-L1 by inhibiting its polyubiquitination by E3 ligase HUWE1

TMUB1 overexpression mitigated the turnover of PD-L1 in the cycloheximide-chased MDA-MB-231 cells, while the opposite

**Fig. 1 | TMUB1 is a positive regulator of PD-L1. a** GSEA analysis of PD-L1 expression (PD-L1 high/PD-L1 low) in the TCGA BRCA dataset. Enriched post-translational modification pathways and enrichment scores are shown. **b**, **c** Relative glycosylation (n = 9) and (**c**) phosphorylation levels of PD-L1 (n = 3) were detected by mass spectrometry in tumor tissue (T) and adjacent non-malignant tissue (N) from breast cancer patients. Two-sided Wilcoxon test for (**b**) and Two-sided Mann–Whitney test for (**c**). **d** Venn Diagram showing IP-MS-detected potential PD-L1-interacting protein, upregulated genes in BRCA (TCGA) and genes associated with tumor CD8+ T cell infiltration in BRCA (TCGA). **e** Representative IHC staining of TMUB1 and CD8 in selected TMUB1-High and TMUB1-Low patients breast cancer tissue from SYSUCC cohorts. Scale bar: 100 μm. **f** Correlations between the TMUB1 positivity rate and CD8 positivity rate in breast cancer tissue sample (n = 55). The R values and p values are from Pearson's correlation analysis. **g** Kaplan-Meier analysis of the overall survival curve for breast cancer patients (SYSUCC cohorts; n = 100) with TMUB1 and

PD-L1 combined expression. Kaplan-Meier analysis along with log-rank test. **h** Co-IP analysis of the interaction between endogenous PD-L1 and endogenous TMUB1 within MDA-MB-231 cells. IgG was used as the negative control. **i**–**k** Flow cytometric analysis with median fluorescence intensity (MFI) of PD-L1 in control, stable TMUB1-Flag expression (**i**) or TMUB1-knockdown MDA-MB-231 cells (**j**) or with the stimulation of IFN-γ (**k**). Data are presented as means ± SEM; n = 7; One-way analysis of variance (ANOVA) followed by Tukey's test. **l** The sh-scramble or sh-TMUB1 MDA-MB-231 cells co-cultured with activated T cells for 48 h and then subjected to crystal violet staining. The ratio of MDA-MB-231 cells to T cells: 1:3. Data are presented as mean ± SEM; n = 4. One-way ANOVA followed by Tukey's test. **m**–**n** The RT-qPCR analysis of the mRNA expressions of IFN-γ (**m**) or TNF-α (**n**) in PBMCs after co-culture with MDA-MB-231 sh-scramble cells or sh-TMUB1 cells. Data are presented as mean ± SEM; n = 4. One-way ANOVA followed by Tukey's test.

phenotype was observed in the TMUB1-knockdown cells (Fig. 2a, b and Supplementary Fig. 3a, b). Previously reported PD-L1-associated degradation pathways mainly include lysosomal and proteasomal pathways[34,37]. Indeed, the proteasomal inhibitor MG132, as well as the lysosomal inhibitor chloroquine, could stabilize PD-L1; however, only MG132 treatment partially rescued the expression of PD-L1 in the TMUB1-knockdown cells (Fig. 2c). In addition, the co-localization between PD-L1 and lysosomes was not affected by the changes of TMUB1 (Supplementary Fig. 3c, d), indicating that TMUB1 regulates the proteasome-dependent degradation of PD-L1.

TMUB1 affects the ubiquitinated proteasomal degradation pathway of multiple proteins by direct binding or indirect regulation[20,21]. TMUB1 strongly blocked the polyubiquitination of endogenous PD-L1 to reduce its degradation (Fig. 2d, e). Strikingly, our mass spectrometry data revealed an E3 ligase, named HUWE1, which has a peptide score close to that of TMUB1, as a binding protein for PD-L1, suggesting that HUWE1 and TMUB1 have comparable binding ability toward PD-L1 (Fig. 2f). HUWE1 is a multifaceted HECT domain-containing ubiquitin E3 ligase, which is associated with tumorigenesis and metastasis[38]. An immunoprecipitation assay confirmed the binding between PD-L1 and HUWE1 (Fig. 2g), while the direct interaction between TMUB1 and HUWE1 was not observed (Supplementary Fig. 3e). HUWE1 overexpression resulted in PD-L1 degradation and knockdown of HUWE1 in the MDA-MB-231 cell line could effectively prolong the half-life of PD-L1 (Supplementary Fig. 3f). Overexpression of TMUB1 counteracted the effect of HUWE1, while TMUB1 depletion exacerbated this effect, suggesting that HUWE1 and TMUB1 antagonistically affect the degradation of PD-L1 (Fig. 2h, i). In addition, overexpression of HUWE1 increased the polyubiquitination of PD-L1, while TMUB1 overexpression reduced it (Fig. 2j). Mutation analysis revealed that K178 and K281 were associated with PD-L1 protein stability and ubiquitination, while only K281, but not K75, K178 or K185, is required for the HUWE1-induced ubiquitination and degradation of PD-L1 (Supplementary Fig. 3g–j). Altogether, these findings demonstrated that TMUB1 stabilized PD-L1 by counteracting its polyubiquitination by HUWE1 and degradation (Fig. 2k).

**TMUB1 drives the posttranslational modifications of PD-L1 against its ER-associated degradation**

Accumulating evidence suggests that TMUB1 was localized on the membrane of the endoplasmic reticulum to function as a member of the endoplasmic reticulum-associated degradation (ERAD) complex and regulate the degradation of ERAD substrates, including PD-L1[18,25]. We revealed that TMUB1 could bind different forms of PD-L1. Moreover, non-glycosylated PD-L1 was induced upon TMUB1 overexpression (Fig. 3a and Supplementary Fig. 4a). It is noteworthy that HUWE1 also bound different forms of PD-L1 upon MG132 treatment (Fig. 3b and Supplementary Fig. 4b). It is known glycosylation plays an important role in regulating PD-L1 stability[9]. To determine the

relationship between TMUB1 and non-glycosylated PD-L1, a 4NQ (N35Q\N192Q\N200Q\N219Q) PD-L1 mutant, whose glycosylation was completely abolished, was constructed (Supplementary Fig. 4c and 4d). Consistent with previous studies, the PD-L1 4NQ mutant underwent faster protein degradation upon CHX treatment (Supplementary Fig. 4e). Notably, TMUB1 and HUWE1 could bind directly to the PD-L1 4NQ mutant (Supplementary Fig. 4f, g) and both are consistent with previous findings in the regulation of the stability of the PD-L1-4NQ mutant (Supplementary Fig. 4h–j). According to these findings, we speculate that HUWE1 and TMUB1 play antagonistic roles for each other to degrade the non-glycosylated PD-L1.

Glycosylation occurs inside the ER and non-glycosylated proteins mainly aggregate inside the ER. Consistent with this, the ER localization of TMUB1 to HUWE1 was validated by several experiments, including immunofluorescence staining, continuous sucrose density gradient centrifugation[39], discontinuous sucrose density gradient centrifugation[40], and differential centrifugation[41] (Fig. 3c–e and Supplementary Fig. 4k, l), suggesting the binding of PD-L1 to both TMUB1 and HUWE1 would occur in the same subcellular organelle (Fig. 3f).

To identify the binding regions of TMUB1 in PD-L1, truncated variants containing extracellular domain and transmembrane and cytoplasmic domain were constructed. Interestingly, TMUB1 could bind to multiple domains of PD-L1, while HUWE1 only bound the extracellular domain of PD-L1 (Fig. 3g–i). According to this finding and our mass spectrometry data, we hypothesized that PD-L1 is protected by TMUB1 from binding to HUWE1; thus, its immature form could escape the ERAD. Indeed, the interaction between HUWE1 and PD-L1 was influenced by TMUB1 and vice versa, suggesting a competitive relationship between TMUB1 and HUWE1 to bind PD-L1 (Fig. 3j and Supplementary Fig. 4m–o). Furthermore, the His-tagged TMUB1 and GST-tagged PD-L1 isolated from *E. coli* were incubated with HUWE1-Flag obtained from the HEK-293T cells and subjected to in vitro pull-down assay, which demonstrated that TMUB1 impaired the binding between HUWE1 and PD-L1 (Supplementary Fig. 4p).

Glycosylation-deficient PD-L1 could be targeted for ERAD degradation[18]. All of the above post-translational modifications of PD-L1 regulated by TMUB1 are directly related to the ERAD-based degradation of PD-L1. Consistent with the above findings, treating cells with Eeyarestatin I (Eer I), an inhibitor of ERAD[42], which could effectively inhibit the ubiquitination degradation pathway of proteins in the endoplasmic reticulum, rescued the PD-L1 protein expression in the TMUB1-knockdown cells (Fig. 3k), suggesting that TMUB1 was involved in the ERAD-based degradation of PD-L1.

However, Eer I treatment, in addition to eliminating the reduction in total PD-L1 caused by TMUB1 knockdown, did not reverse the diminished PD-L1 glycosylation. Further, we completely blocked the intracellular degradation pathway of PD-L1 with CQ and MG132 and still obtained similar results, suggesting that TMUB1 may regulate glycosylation of PD-L1 (Fig. 3l). The proteomics data from the CPTAC BRCA database by Linkedomics (http://linkedomics.org) showed that STT3A

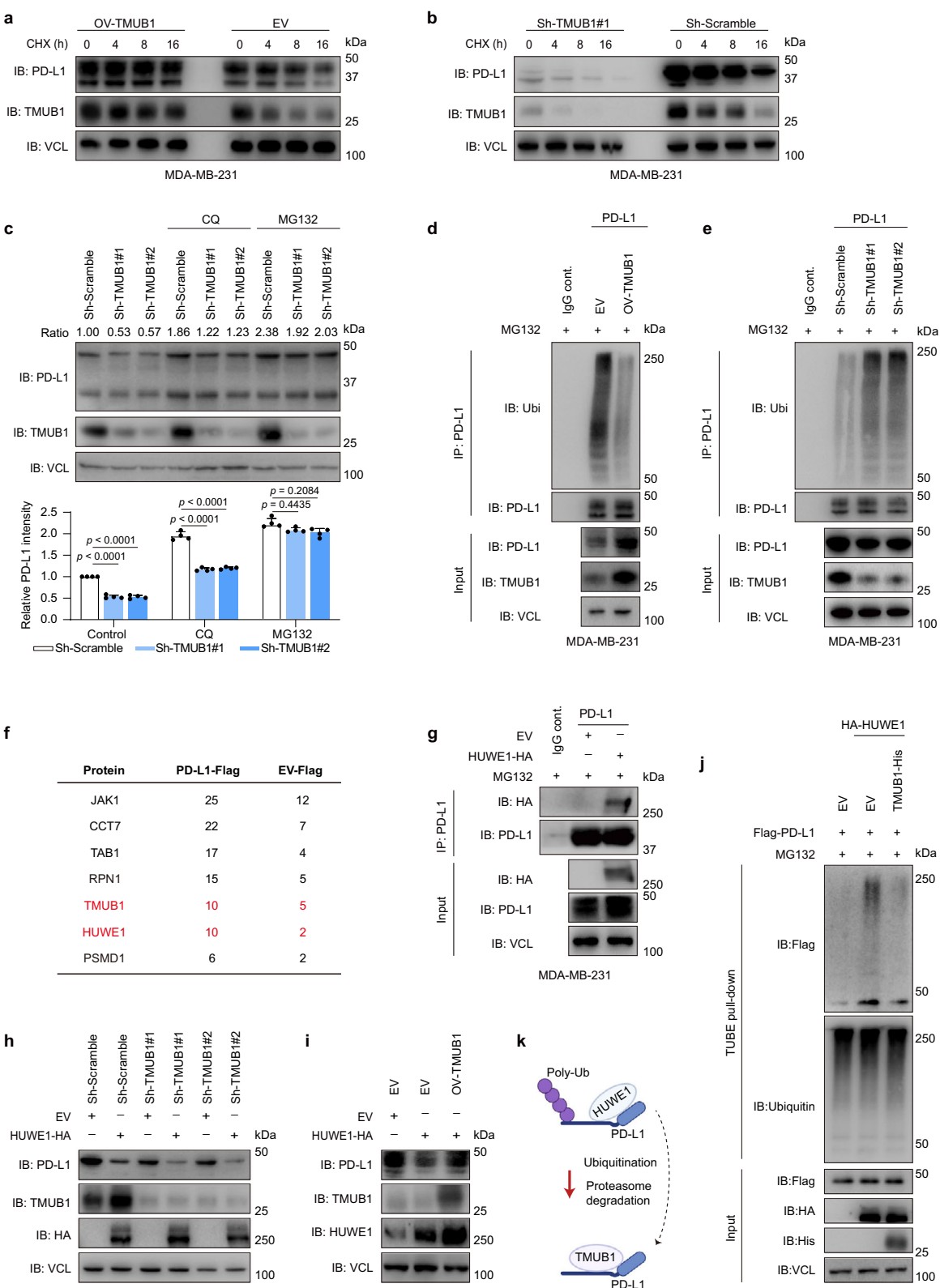

is one of the most significantly associated proteins of TMUB1 (Fig. 3m) and the GSEA analysis of the TMUB1 associated proteins showed that TMUB1 was closely associated with the glycosylation process (Supplementary Fig. 5a)[43]. STT3A is the catalytic subunit of the oligosaccharyltransferase (OST) complex, which catalyzes the N-glycosylation of PD-L1[28]. We found that both TMUB1 and PD-L1 interacted with STT3A and TMUB1 overexpression enhanced the

interaction between PD-L1 and STT3A (Fig. 3n and Supplementary Fig. 5b, c). Together, TMUB1 promotes PD-L1 glycosylation by enhancing the binding of PD-L1 to STT3A.

These findings together indicated a regulatory mechanism for PD-L1: both TMUB1 and HUWE1 are involved in the ER-associated degradation of PD-L1, where TMUB1 protected PD-L1 from ERAD to promote the glycosylation of PD-L1 for improving PD-L1 stability.

**Fig. 2 | TMUB1 stabilizes PD-L1 by antagonizing its polyubiquitination by the E3 ligase HUWE1. a, b** Immunoblots of PD-L1 in MDA-MB-231 cells with stable over-expression of Tag-free TMUB1 (**a**), TMUB1-knockdown (**b**) or control following treatment with 20 μg/mL cycloheximide (CHX) for the indicated time points. **c** Immunoblots of PD-L1 in TMUB1-knockdown or control MDA-MB-231 cells following treatment with 10 μM of MG132 or 50 μM of CQ for 6 h. Data are presented as mean ± SEM; n = 4. One-way ANOVA followed by Tukey's test. Co-IP analysis of the interaction between endogenous ubiquitin and PD-L1 in stable Tag-free TMUB1-overexpression (**d**), TMUB1-knockdown (**e**) or control MDA-MB-231 cells treated with 10 μM of MG132 for 6 h. **f** The MS analysis to explore different PD-L1-binding proteins in Fig.1d. Numbers represent the peptide-spectrum matches of different proteins detected by mass spectrometry in the indicated groupings. The representative candidates are listed. TMUB1 and HUWE1 were identified. **g** Co-IP analysis of the interaction between endogenous PD-L1 and HUWE1-HA in MDA-MB-231 cells treated with 10 μM of MG132 for 6 h. **h, i** Immunoblots of PD-L1 in TMUB1-knockdown (**h**), Tag-free TMUB1-overexpression (**i**) or control MDA-MB-231 cells transfected with HA-empty vector or HA-HUWE1. **j** TUBE-pull down assay of the interaction between PD-L1-Flag and endogenous ubiquitin in stable TMUB1-His or empty-vector overexpression MDA-MB-231 cells transfected with HA-HUWE1 or HA-empty vector. **k** The working model of TMUB1 antagonizing HUWE1-mediated PD-L1 ubiquitination and enhancing PD-L1 stability.

## TMUB1 knockdown promotes antitumor immunity in vivo through PD-L1 degradation

To evaluate the role of TMUB1 in PD-L1 degradation and tumorigenesis, EO771, 4T1 and MFC cell lines with stable Tmub1 knockdown were established. Mouse TMUB1 and human TMUB1 are highly conserved with a few amino acid sequence homology differences. Their regulatory effects on PD-L1 are similar (Fig. 4a). Change of TMUB1 protein level did not affect the cell proliferation for human breast cancer MDA-MB-231 cells, mouse breast cancer 4T1, EO771 cells, and gastric cancer MFC cells (Supplementary Figs. 6a–d, 7a). Moreover, the 4T1 cells with Tmub1 knockdown did not show any differences in tumor growth in immunodeficient nude mice compared to the control mice (Fig. 4b and Supplementary Fig. 6e, f). However, reduced EO771 and 4T1 tumor growth was observed in the Tmub1-knockdown group in the immunocompetent C57LB/6 and Balb/c mice (Fig. 4c–g). In line with the mechanistic findings, the IHC and flow cytometry analyses of the tumor tissues revealed that Tmub1-knockdown significantly reduced the PD-L1 levels in EO771 tumor cells (Fig. 4h, j). Consistent with this, mice in the Tmub1-knockdown group exhibited longer survival compared to the control group mice (Fig. 4i). Furthermore, the knockdown of Tmub1 also had a significant effect on the tumor immune microenvironment. Both tumor-infiltrating myeloid-derived suppressor cells (MDSCs) and T$_{reg}$ cells, which play an immunosuppressive role in the tumor microenvironment, showed a certain degree of decrease, accompanied by an increase in the proportion of CD4$^+$ T cells (Fig. 4k–o). Most importantly, the levels of total and activated CD8$^+$ cytotoxic T cells (GzmB$^+$) that infiltrated the tumor microenvironment increased, while the proportion of exhausted CD8$^+$ T cells decreased, indicating that the deficiency of Tmub1 maintains high cytotoxic T lymphocyte activity (Fig. 4p–r). In addition, the expressions of inflammatory cytokines and T cell chemokines, including IFN-γ, TNFα, CCL-5, and CXCL-10, increased with Tmub1 knockdown (Fig. 4s–w). This observation indicated that the loss of Tmub1 in tumor cells disrupted the immunosuppressive tumor immune microenvironment and led to the enhanced antitumor immune response. We similarly validated our findings on the gastric cancer cell line MFC and the more malignant breast cancer cell line 4T1 tumor-bearing Balb/c mice (Supplementary Fig. 6g–q and Supplementary Fig. 7). These findings demonstrated that TMUB1 plays an important role in the regulation of PD-L1 for the development of antitumor immunity.

## Clinical implication of TMUB1 as a potential target in tumor immunotherapy

To evaluate the clinical roles of the TMUB1-mediated regulation of PD-L1, we assessed the levels of TMUB1 and PD-L1 in the tissue samples from the tissue microarrays integrated from individuals with breast cancer, which were obtained from the Cancer Center of Sun Yat-sen University (Supplementary Table 3), and from individuals with gastric cancer, which were obtained from the Second Affiliated Hospital of Zhejiang University (Supplementary Table 4). The IHC analysis revealed that tumors with high TMUB1 levels showed a high level of PD-L1, which showed a statistically significant positive correlation between them based on the expression score (Fig. 5a, b). The participants were categorized into two separate groups-TMUB1-high/low and PD-L1-high/low

groups-based on the levels of TMUB1/PD-L1 compared to the respective median value for all individuals. It was observed that individuals in the TMUB1-high group almost overlapped with those in the PD-L1-high group (Fig. 5c). Moreover, upregulation of both PD-L1 and TMUB1 was observed in breast cancer tumors compared to the corresponding adjacent non-malignant tissues (Fig. 5d–g). Further, TMUB1/STT3A/HUWE1 expression in breast cancer tumors was assessed using RT-qPCR. The participants were categorized into two groups based on the gene expression, and it was revealed that low TMUB1/STT3A levels benefited the overall survival rate in the individuals with breast carcinoma, while HUWE1 is the opposite, consistent with our previous findings in cells. (Fig. 5h and Supplementary Fig. 9a, b). A similar series of analyses were performed for the individuals with gastric cancer, and the same conclusion was reached, *i.e.*, higher TMUB1 levels were associated with higher PD-L1 levels and worse clinical outcomes in the patients with gastric carcinoma (Fig. 5i–p and Supplementary Fig. 9c, d). These findings were consistent with the discovery that TMUB1 promotes PD-L1 degradation, suggesting that the increase in the TMUB1 expression in cancer results in PD-L1 accumulation that drives the escape of cancer/tumor cells from antitumor immunity. Collectively, these results suggest that the TMUB1–PD-L1 axis is involved in breast cancer and gastric cancer tumorigenesis, highlighting the potential of utilizing this axis as a therapeutic target in the treatment of breast and gastric cancers.

## The competitive peptide PTPR weakened the upregulation of PD-L1 by TMUB1

Given the important role of TMUB1 in PD-L1 regulation and its potential clinical significance, therapeutic strategies based on TMUB1 were explored. A range of drugs targeting either PD-L1 or its upstream regulatory molecules is available currently[29,44]. In addition to screening the existing drugs for this objective, designing small-molecule inhibitors based on binding sites or binding domains is another effective option commonly applied[45,46]. Blockers of the post-translational modification sites of PD-L1 have shown clinical translation potential[10]. Given these facts, small molecules were designed to block the binding of PD-L1 to TMUB1.

Since several domains in the PD-L1 protein can bind TMUB1 (Fig. 3i), different truncated variants of TMUB1 were constructed, which was followed by a co-immunoprecipitation assay to verify the binding of TMUB1 to PD-L1 and the associated functional domain (Fig. 6a). Loss of the MU1 domain (32–102), which is highly conserved in mammals, prevented the association of TMUB1 with PD-L1, while the nucleus-cytoplasm shuttle ability remained unchanged (Fig. 6b and Supplementary Fig. 10a, b). This result was corroborated in the TMUB1-knockout cell lines. The reconstitution of wild-type TMUB1 could rescue the degradation of PD-L1, while the MU1-deleted TMUB1 failed to do so (Supplementary Fig. 10c). Furthermore, it was narrowed down to the C-terminus of the MU1 domain, in the region between amino acids 88 and 102, which was designated as the PR (PD-L1-regulating) domain (Fig. 6c). The absence of this domain in TMUB1 resulted in a significant decrease in its ability to bind and regulate PD-L1 (Fig. 6d).

Given the short amino acid sequence length and the electrical suitability, it was speculated that the peptide of this 15AA could achieve

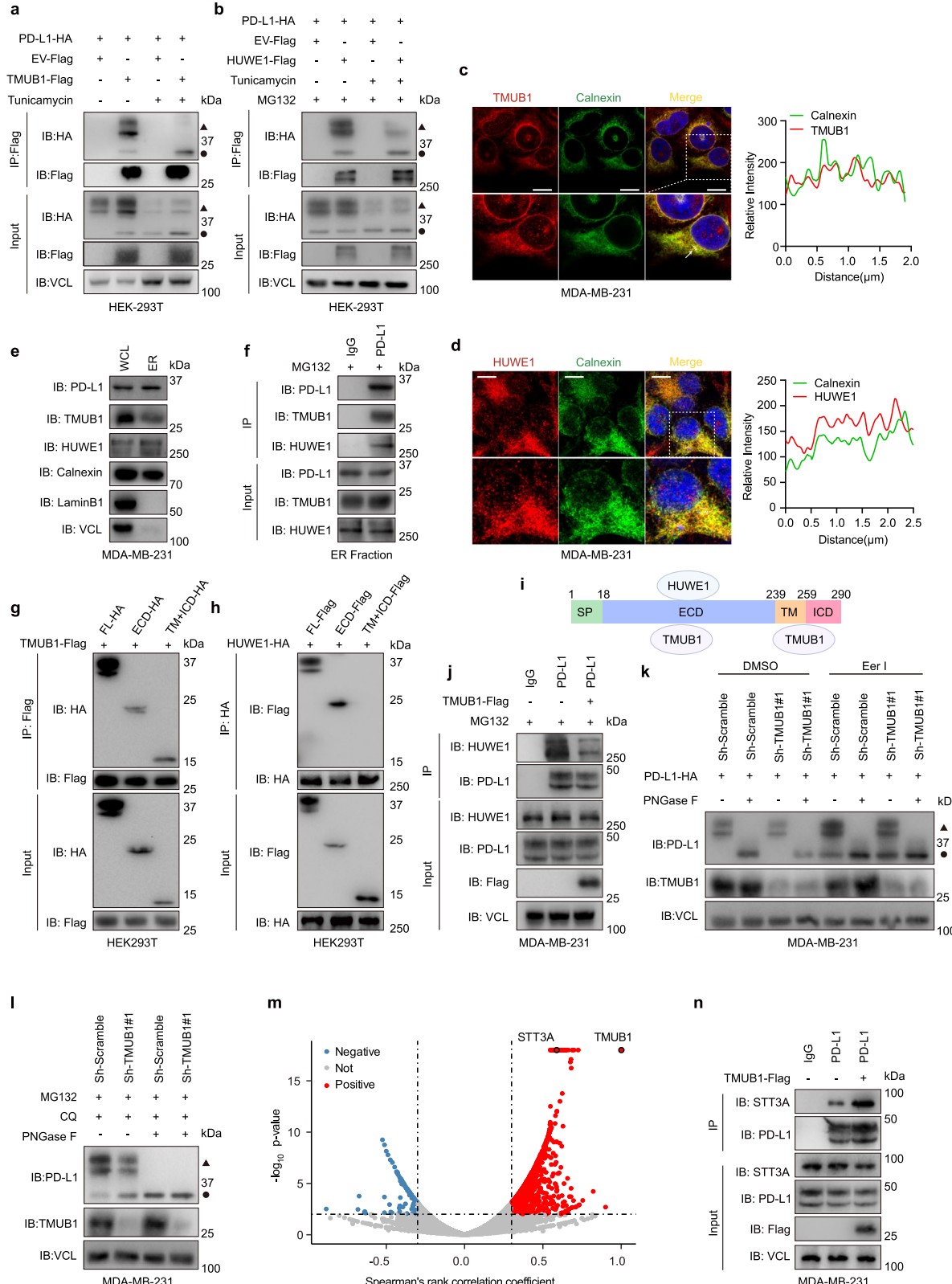

intracellular delivery without the cell-penetrating peptide. Therefore, a peptide of the FITC-Tagged PR domain was developed and designated as PTPR (Peptide of TMUB1 PD-L1 Regulatory domain) (Fig. 6e). It was identified using the immunofluorescence analysis that this peptide can penetrate the cell membrane and enter the cell (Fig. 6f). Treatment of MDA-MD-231 with a concentration gradient of PTPR revealed 10 µM as the appropriate working concentration for PTPR (Supplementary

Fig. 9d). Indeed, the addition of PTPR weakened the binding between PD-L1 and TMUB1, enhanced the binding of HUWE1 to PD-L1, which in turn promotes its ubiquitinated degradation and reduced the level of PD-L1 protein caused by TMUB1 overexpression (Fig. 6g–k). Furthermore, treatment with 10 µM PTPR for 12 h did not result in changes in protein abundance of HUWE1 and RNA levels of TMUB1\HUWE1\PD-L1 (Supplementary Fig. 10e, f). Moreover, PTPR enhanced the killing of

**Fig. 3 | TMUB1 protects PD-L1 from HUWE1-mediated ER-associated degradation. a, b** Co-IP analysis and immunoblots for PD-L1-HA and TMUB1-Flag (**a**) or HUWE1-Flag (**b**) in the HEK-293T cells treated with 5 μg/ml Tunicamycin for 12 h or not or 10 μM of MG132 for 6 h. Triangle: Glycosylated PD-L1, Circle: Non-glycosylated PD-L1. **c, d** Endoplasmic reticulum localization of TMUB1 (**c**) and HUWE1 (**d**) was detected using immunofluorescence staining (left), Calnexin was stained to characterize the endoplasmic reticulum. Line scan of the relative fluorescence intensity of the signal (dotted line; left), demonstrating peak overlapping (right). Scale bar: 10 μm. **e** Analysis of the subcellular localization of PD-L1, TMUB1, and HUWE1 in the fraction of MDA-MB-231 cells isolated by differential centrifugation. **f** Co-IP analysis for PD-L1 and TMUB1/HUWE1 in the ER fraction of MDA-MB-231 cells. **g** Co-IP analysis for the interaction of the different truncations of PD-L1-HA with TMUB1-Flag in HEK-293T cells. **h** Co-IP analysis for the interaction of the different truncations of PD-L1-Flag with HUWE1-HA in HEK-293T cells. **i** Schematic diagram for PD-L1 displaying the position of the Extracellular Domain (ECD) and the Transmembrane and Cytoplasmic Domain (T + C). **j** Co-IP analysis of the interaction between endogenous HUWE1 and PD-L1 within MDA-MB-231 cells with TMUB1 overexpression or not. IgG was used as the negative control. **k** Immunoblots of PD-L1 in TMUB1-knockdown or control MDA-MB-231 cells following treatment with 10 μM of Eer I for 12 h. Cell lysates were treated with PNGase F. Triangle: Glycosylated PD-L1, Circle: Non-glycosylated PD-L1. **l** Immunoblots of PD-L1 in TMUB1-knockdown or control MDA-MB-231 cells following treatment with 10 μM of MG132 and 50 μM of CQ for 6 h. Cell lysates were treated with PNGase F. Triangle: Glycosylated PD-L1, Circle: Non-glycosylated PD-L1. **m** Volcano plot represented the gene correlation with TMUB1 in breast cancer, p-value < 0.01 and absolute correlation coefficient >0.3 or <−0.3 by using Proteome data of CPTAC - Breast invasive carcinoma prospective cohort from Linkedomics. Two-sided t test. **n** Co-IP analysis for the interaction between endogenous PD-L1 and endogenous STT3A within MDA-MB-231 cells with TMUB1-Overexpression or not. IgG was used as the negative control.

the tumor cells by T cells, as well as the expression levels of TNF-α and IFN-γ in T cells (Fig. 6m–o). These results indicated that PTPR effectively inhibits the interaction between PD-L1 and TMUB1 and downregulates the cellular abundance of PD-L1 (Fig. 6l).

Our finding revealed that PTPR exhibits significant PD-L1 inhibition on cells. Further, we evaluated the in vivo antitumor efficacy of PTPR. Given the fact that PTPR can penetrate the cell membrane, we adopted the experimental approach of direct intratumoral injection of PTPR at 10 mg·kg⁻¹ into Balb/c or C57BL/6 mice injected with 4T1, EO771, or MFC cells (Fig. 7a and Supplementary Fig 11a, 12a). Tumor growth was significantly inhibited and life span was increased in the mice injected with PTPR compared to the vehicle-only group (Fig. 7b–e). Moreover, PTPR-injection led to downregulation of PD-L1 in tumors and changes in the immune microenvironment of tumors, especially the upregulation of total and activated CD8⁺ cytotoxic T cells levels in tumor-infiltrating lymphocytes (Fig. 7f–i and Supplementary Fig. 13a–f). Also consistent with our previous findings, the use of antibodies to deplete the immune microenvironment of CD8⁺ T cells rather than CD4⁺ T cells or NK cells had an impact on the immune efficacy of PTPR (Supplementary Fig. 13g). PTPR also showed promising efficacy in the MFC mouse gastric cancer cells and the 4T1 mouse breast cancer cells (Supplementary Fig. 11 and Supplementary Fig. 12).In addition, PD-L1 knockout also leads to PTPR failure (Supplementary Fig. 13h, i).

Some recent studies have revealed that combining multiple immune checkpoint therapies in immunotherapy could improve efficiency[47]. PTPR was used to restrict PD-L1 in tumor tissue by targeting TMUB1, and an αCTLA4 treatment was performed in the presence/absence of PTPR in EO771 tumor-bearing C57BL/6 mice (Fig. 7j). Tumor growth was significantly decreased both by the PTPR treatment and the αCTLA4 treatment. Of interest, the combination of the two immunotherapies achieved better efficacy, a further decrease of tumor growth, and even complete regression and prolonged survival in the tumor-bearing mice were observed, which is attributed to the increased level of activated CD8⁺ cytotoxic T cells (GzmB⁺) infiltration in the tumors made by the combination therapy (Fig. 7k–p). Taken together, our data indicate that PTPR is a promising agent to boost antitumor immunity and αCTLA4 immune checkpoint blockade immunotherapies.

In addition to the effectiveness of PTPR, we characterized the in vivo toxicity of PTPR by challenging Balb/c mice with PTPR at 0, 100, 200, and 500 mg·kg⁻¹ (n = 3 for each group) (Supplementary Fig. 13j). In the 500 mg·kg⁻¹ group, the injected peptide was equivalent to 1/2,000 body weight, equivalent to a very high dose. The survival, vitality, and gross appearance of the mice were monitored regularly, and hematoxylin and eosin (H&E) staining was performed in different organs at the end of the experiment (two week after injection). We found that 9 of the 9 mice injected with 100 mg·kg⁻¹ or higher were alive at the endpoint, and there was no apparent difference in the gross appearance and behavior between the different groups (Supplementary Fig. 13k). Furthermore, only the lung exhibited damage at the 200 mg·kg⁻¹ and 500 mg·kg⁻¹ dose, without any abnormality in the other organs (liver, heart, stomach, colon, stomach, kidney, spleen, and brain) (Supplementary Fig. 13l). In addition, these H&E staining results did not indicate drug-induced liver injury. These results suggest that, although high-dose PTPR resulted in some toxicity in the lung, this toxicity was well-tolerated and all of the tested animals survived until the end of the experiment.

## Discussion

The strategy of immune checkpoint blockade by anti-PD-L1/PD-1 has exhibited unprecedented clinical efficacy, although it presents a low response rate, which has been attributed closely to the level of PD-L1 expression in tumor cells. Therefore, deciphering the molecular mechanisms underlying the regulation of PD-L1 expression is of great significance. Recent studies have reported different post-translational modification mechanisms for regulating PD-L1 protein expression, including glycosylation and ubiquitination; however, regulation of PD-L1 stabilization by the ubiquitin-proteasome system has not been completely understood yet[9,28,48].

Our work reveals the mechanism by which TMUB1 and HUWE1 regulate PD-L1 stability through post-translational modifications. Both TMUB1 and HUWE1 are localized in the endoplasmic reticulum and are involved in the ERAD of PD-L1 (Fig. 7q). A recent study reported that deubiquitinase OTUB1 maintains the newly-synthesized PD-L1 proteins in a less ubiquitinated state to save them from ERAD degradation[49]. Furthermore, our study indicated that TMUB1 might be involved in a more prior process, preventing PD-L1 ubiquitination rather than leading to PD-L1 deubiquitination. TMUB1 was found to bind non-glycosylated PD-L1 in the endoplasmic reticulum and recruit STT3A to promote glycosylation of PD-L1, resulting in its protection from being ubiquitinated upon HUWE1 binding and ultimately increasing the levels of PD-L1 in different cancer types.

Notably, a growing number of studies have focused on roles other than PD-L1 as a membrane ligand, including transcriptional control in the nucleus and participation in the activation of other signaling pathways[27,50–52]. The complicated intracellular regulatory roles of PD-L1 imply that we should concentrate on the mechanisms that determine the total abundance of tumor cell-intrinsic PD-L1, such as PD-L1 regulation via TMUB1.

As a functional element localized in the ER regulating the stability of a wide range of proteins, the importance of TMUB1 may be greater than previously assumed. This also renders it essential to elucidate the regulatory mechanisms and related upstream biology events and molecules associated with TMUB1, including the identification of its nucleoplasmic shuttle mechanism and subsequently targeting these in combination with relevant clinical therapies. Recent studies have demonstrated that the ketogenic diet regulates PD-L1 through the

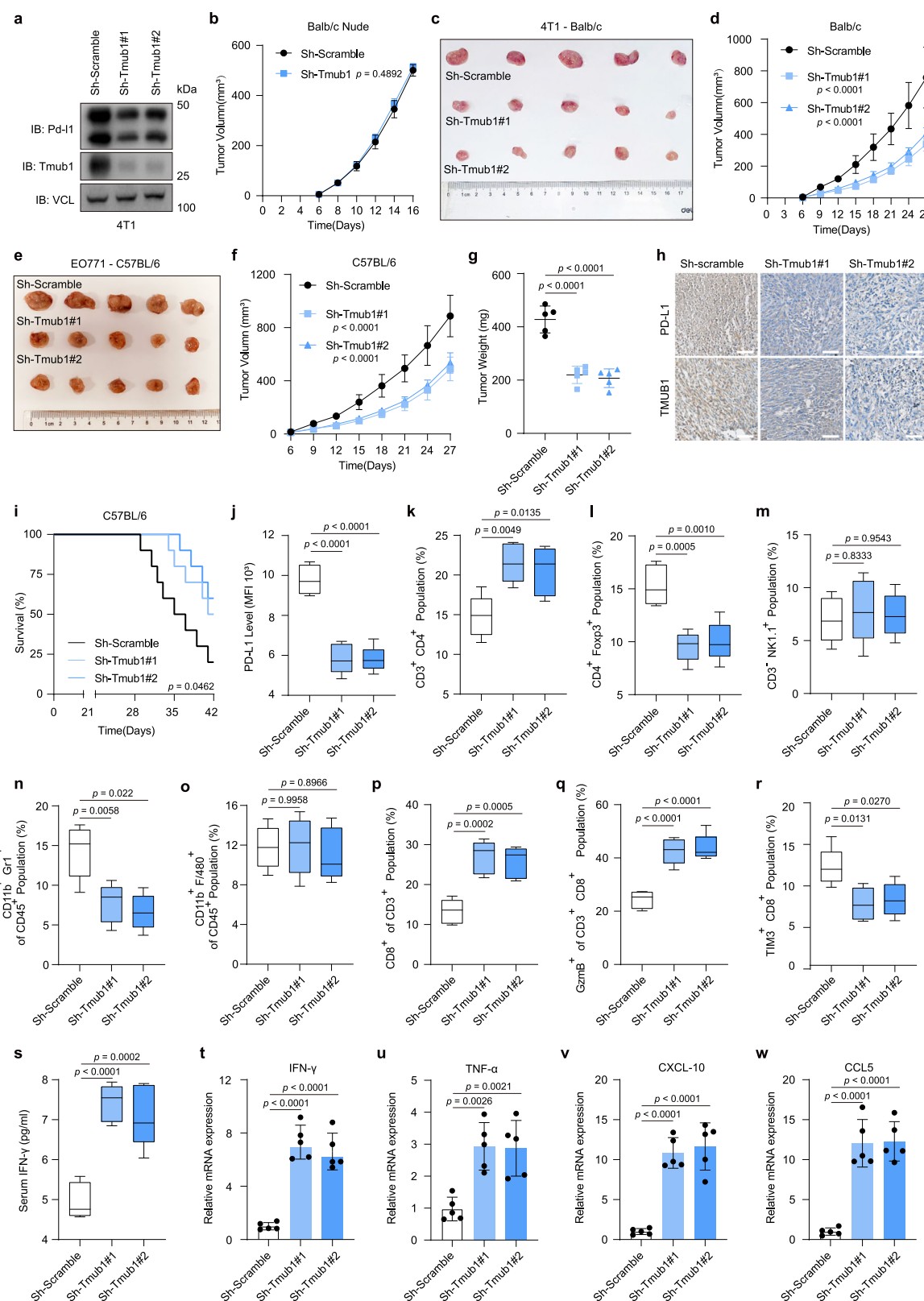

activation of AMPK[53], while the endoplasmic reticulum is an important site of lipid metabolism and TMUB1 is involved in the regulation of HMG-CoA reductase[54], suggesting TMUB1 might be a regulator in the metabolic-immune axis. In addition, it has been revealed that TMUB1 regulates NF-κB pathway in some cells[21]. Although TMUB1 does not appear to serve the same role in breast cancer cells due to differences in cell type or tumor selectivity. Targeting TMUB1 is anticipated to

achieve dual restriction of PD-L1 RNA and protein levels, hence enhancing the significance of TMUB1 regulation and its research and clinical value.

In a previous study, short peptides synthesized for PD-L1-targeting could competitively inhibit PD-L1 palmitoylation[10]. Similarly, the amino acid sequence of TMUB1 was resolved, and the functional domain that binds and regulates PD-L1 was identified. Based on

**Fig. 4 | TMUB1 knockdown promotes antitumor immunity in vivo via PD-L1 degradation. a** Immunoblots of PD-L1 and TMUB1 in control or Tmub1-knockdown 4T1 cells. **b** Analysis of 4T1 cell tumor growth in the xenograft Balb/c nude mouse model. Data are presented as mean ± SEM of n = 5 mice per group. Two-way ANOVA. **c–f** Analysis of tumor growth in the xenograft mouse model established using indicated 4T1 cells in Balb/c mice (**c**, **d**) or indicated EO771 cells in C57BL/6 mice (**e**, **f**). In vivo generated tumors are depicted. Data are presented as mean ± SEM. of n = 5 mice per group. Two-way ANOVA. **g** Analysis of tumor weight in the xenograft mouse model. Data are presented as mean ± SEM of n = 5 mice per group. One-way ANOVA followed by Tukey's test. **h** Representative IHC staining in randomly selected tumors from mice subcutaneously injected with the indicated stably-transduced EO771 cells. Scale bar: 100 μm. **i** Survival in the mice bearing sh-Scramble or sh-Tmub1 EO771-derived tumor. n = 10 mice per group. Log-rank test.

**j–s** Flow cytometric analysis with median fluorescence intensity (MFI) of PD-L1, (**j**) CD3$^+$ CD4$^+$ T cells (**k**), CD4$^+$ Foxp3$^+$ T$_{reg}$ cells (**l**), CD3$^-$ NK1.1$^+$ NK cells (**m**), CD11b$^+$ Gr1$^+$ MDSC (**n**), CD11b$^+$ F/480$^+$ TAM (**o**), CD3$^+$ CD8$^+$ T cells (**p**), granzyme B-positive CD3$^+$ CD8$^+$ T cells (**q**), TIM3$^+$ CD8$^+$ T$_{Exhausted}$ cells (**r**) and the abundance of IFN-γ (pg/mL) in mice serum was detected by ELISA assays (**s**). Data are presented as a box plot with box and whiskers. Bounds of box show the 25th and 75th percentiles, and the central lines in the box represent the median value. Whiskers show min to max value, n = 5 per group. One-way ANOVA followed by Tukey test. Detailed gating information is presented in Supplementary Fig. 8. **t–w** The RT-qPCR analysis of the expressions of IFN-γ (**t**), TNF-α (**u**), CXCL-10 (**v**), and CCL-5 (**w**) in bulk EO771 tumor xenografts. Data presented as mean ± SEM; n = 5. One-way ANOVA followed by Tukey's test.

this identified domain, a peptide named PTPR was designed and synthesized, which could competitively bind to PD-L1 to weaken the regulatory effect of TMUB1 at the cellular level. We have conducted a preliminary assessment of the efficacy and toxicity of PTPR both in vitro and in vivo. Mice with tumors treated with a combination of PTPR and αCTLA-4 yielded significant outcomes, suggesting a prospective application. As an artificially designed molecule, of course, there is much more to be studied in depth, including compound modification, delivery technology development, and the functional mechanisms affecting the binding of TMUB1 and HUWE1 to PD-L1.Whether this involves more complex regulatory networks, including structural resolution of the complexes, remains to be determined by further research.

In summary, our study revealed two regulators of PD-L1, namely, TMUB1 and HUWE1, and connected TMUB1 with immune evasion by cancer cells. Furthermore, we showed that TMUB1 induced PD-L1 stability by regulating the post-translational modifications of PD-L1, protecting it from binding to HUWE1, and recruiting STT3A to promote PD-L1 glycosylation, thereby saving it from ERAD degradation, and promoting tumor growth in vivo by facilitating PD-L1-mediated immune evasion. To this end, we have developed a peptide targeting TMUB1 and have achieved significant therapeutic results in tumor-bearing mice. In conclusion, TMUB1 is a potential candidate target that could be utilized to improve patient outcomes in cancer immune check blockade therapy owing to its influence on the PD-L1 level in the tumor microenvironment.

## Methods

### Contact for reagent and resource sharing
Further information and requests for resources and reagents should be directed to and will be fulfilled by the Lead Contact, Aifu Lin (linaifu@zju.edu.cn).

### Cell lines
The human breast cancer cell lines MDA-MB-231 (CRM-HTB-26; RRID: CVCL_0062) and MDA-MB-468 (HTB-132; RRID: CVCL_0419), the human embryonic kidney cell line HEK-293T (CRL-3216; RRID: CVCL_0063), the human lung cancer cell line A549 (CCL-185; RRID: CVCL_0023), the human liver cancer cell line HepG2 (HB-8065; RRID: CVCL_0027), the human gastric cancer cell line AGS (CRL-1739; RRID: CVCL_0139), the mouse breast cancer cell line 4T1 (CRL-2539; RRID: CVCL_0125) the mouse gastric cancer cell line MFC (RRID: CVCL_5J48), and the mouse breast cancer cell line EO771 (CRL-3461; RRID: CVCL_GR23) were purchased from the National Collection of Authenticated Cell Cultures (China). All cells were tested for mycoplasma contamination and authenticated using short tandem repeat fingerprinting before use.

### Tissue samples
A total of 100 patients with complete clinicopathologic characteristics and follow-up data who underwent surgery at the Sun Yat-sen University Cancer Center and were histologically diagnosed with breast cancer were enrolled. The study protocol was approved by the Institutional Review Board of Sun Yat-sen University Cancer Center. Participants were recruited from Sun Yat-sen University Cancer Center with no perceived bias, and all eligible participants were offered enrollment. Fifty-five tissue samples with complete tissue form were used for IHC staining. All enrolled patients provided written informed consent prior to sample collection. None of the patients were treated with adjuvant radiotherapy or chemotherapy before surgery. Detailed clinical information is listed in Supplementary Table 3.

In addition, 90 patients with complete clinicopathologic characteristics and follow-up data who underwent surgery at the Second Affiliated Hospital, School of Medicine Zhejiang University, and were histologically diagnosed with gastric cancer were enrolled. Written informed consent was obtained from all the patients. Histological cancer types were evaluated by two independent pathologists using the TNM staging guide (2016) released by The American Joint Committee on Cancer (AJCC). Tissue microarrays were made from paraffin-embedded consecutive sections. IHC staining was performed on 72 out of 90 tissue samples. The patients provided written informed written before obtaining study specimens. The experiments were approved by the Ethics Committee of the Second Affiliated Hospital, School of Medicine Zhejiang University. Detailed clinical information is listed in Supplementary Table 4.

### Mice
All animal experiments were performed in accordance with a protocol approved by the Institutional Animal Care and Use Committee, and the mice had a maximum tumor size/burden of less than 20 mm. Experimental protocols were approved by the Animal Care and Use Committee of Zhejiang University School of Medicine (ZJU20210045).

Female mice (Balb/c, Balb/c nude or C57BL/6 J strain; 4–5 weeks old) were purchased from the Shanghai Laboratory Animals Center and used in the xenograft mouse model assay. Animals were housed in a pathogen-free barrier environment (around 20 °C with 40% humidity and 12-h dark/light cycle) throughout the study. Mice were fed a normal chow diet and water with *ad libitum* feeding. Control and experimental animals were bred separately.

### Cloning procedure
PCR was used to clone the full-length PD-L1 and TMUB1 from HEK-293T cDNA. The HUWE1 full-length template was kindly provided by Z.G. Shao (Peking University). All eukaryotic overexpression genes (WT and mutants) were cloned using ClonExpress II One Step Cloning Kit (Vazyme) into pcDNA3.1-Flag/Myc/HA-His, pcDNA3.1-Tag-free or pLVX-SFB empty vectors. By overlapping PCR, all single-point and deletion mutations were created. Cloning into a pMBP28a vector yielded bacterial expression vectors for MBP–His-tagged TMUB1, and cloning into a PGEX-4T-2 vector yielded bacterial expression vectors for GST-tagged PD-L1.

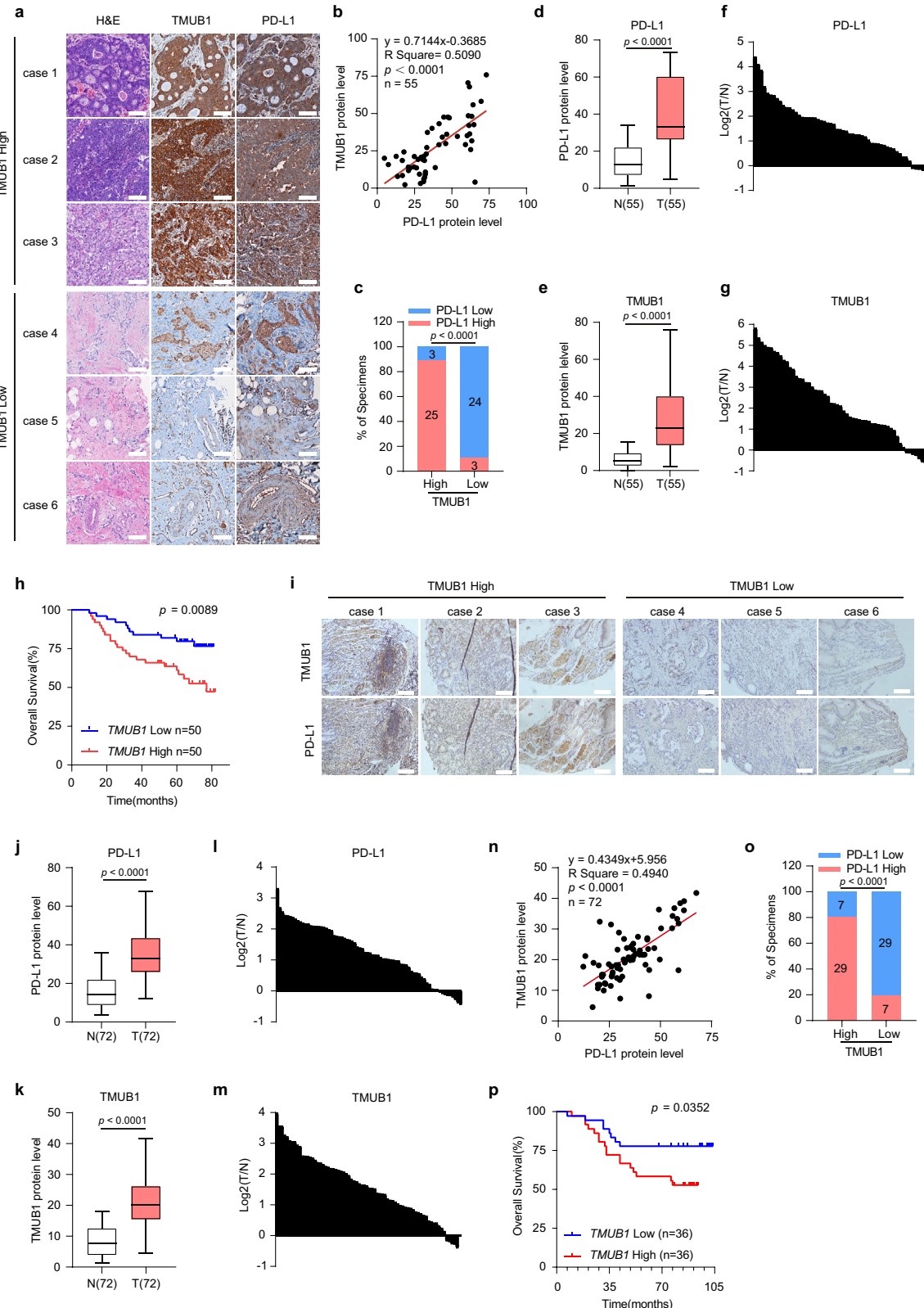

## Short hairpin RNA-mediated knockdown

All short hairpin RNA (shRNA) sequences were designed according to https://portals.broadinstitute.org/gpp/public/. All shRNA sequences were cloned into a pLKO.1-Puro vector by ClonExpress II One Step Cloning Kit (Vazyme). In following functional experiments, the two shRNAs with the highest knockdown efficiency were used. The shRNA sequences used are listed in Supplementary Table 5.

## Data analysis

Raw counts of invasive breast cancer (BRCA) from TCGA were downloaded from the UCSC Xena database (https://xenabrowser.net/datapages/?dataset=TCGA-BRCA.htseq_counts.tsv&host=https%3A%2F%2Fgdc.xenahubs.net&removeHub=https%3A%2F%2Fxena.treehouse.gi.ucsc.edu%3A443), differentially expressed RNAs between tumor and non-malignant samples from breast cancer were analyzed by DESeq2

**Fig. 5 | Clinical value of TMUB1 as a potential target in tumor immunotherapy. a** The expressions of PD-L1 and TMUB1 in 55 primary human breast cancer specimens. Scale bar: 100 μm. **b** Correlations between TMUB1 positivity rate and PD-L1 positivity rate in breast cancer tissues (n = 55). The R values and p values are from Pearson's correlation analysis. **c** Percentages of specimens exhibiting low or high TMUB1 expression were correlated with PD-L1 levels. Two-sided $\chi^2$ test. **d, e** PD-L1 (**d**) and TMUB1 (**e**) protein level in adjacent non-malignant tissue (N) and malignant breast cancer (T). Data are presented as a box plot with box and whiskers. Bounds of box show the 25th and 75th percentiles, and the central lines in the box represent the median value. Whiskers show min to max value, n = 55 per group. Two-sided Wilcoxon test. **f, g** The relative levels of PD-L1 (**f**) and TMUB1 (**g**) in the tumor are normalized to paired non-malignant tissue as differential expression values (T/N). **h** Overall survival curve for breast cancer patients (n = 100) with low or high TMUB1 expression. Kaplan-Meier analysis along with the log-rank test. **i** The expressions of PD-L1 and TMUB1 in 72 primary human gastric cancer specimens. Scale bar: 100 μm. **j, k** PD-L1 (**j**) and TMUB1 (**k**) protein levels in adjacent non-malignant tissue (N) and malignant gastric cancer (T). Data are presented as a box plot with box and whiskers. Bounds of box show the 25th and 75th percentiles, and the central lines in the box represent the median value. Whiskers show min to max value, n = 72 per group. Two-sided Wilcoxon test. **l, m** The relative levels of PD-L1 (**l**) and TMUB1 (**m**) in the tumor are normalized to paired non-tumor tissue as differential expression values (T/N). **n** Correlations between the TMUB1 positivity rate and PD-L1 positivity rate in gastric cancer tissues (n = 72). The R values and p values are from Pearson's correlation analysis. **o** The percentage of specimens exhibiting low or high TMUB1 expression was correlated to PD-L1 levels, Two-sided $\chi^2$ test. **p** Overall survival curve for gastric cancer patients (n = 72) with low or high TMUB1 expression. Kaplan-Meier analysis along with the log-rank test.

(R package (1.16.1)). Genes with adjusted *p*-value < 0.05 and log₂Fold-Change > 0.9 were considered as significantly upregulated expressed.

CD8⁺ T cell infiltration score was downloaded from TIMER 2.0 (http://timer.cistrome.org/). Genes with *p*-value < 0.05 and spearman's rank correlation coefficient ≤ −0.3 were considered as negatively correlated with CD8⁺ T cell infiltration. Detailed sequences are listed in Supplementary Table 2.

FPKM data of BRCA in TCGA was downloaded from the UCSC Data Center (https://xenabrowser.net/datapages/?dataset=TCGA-BRCA.htseq_fpkm.tsv&host=https%3A%2F%2Fgdc.xenahubs.net&removeHub=https%3A%2F%2Fxena.treehouse.gi.ucsc.edu%3A443). After annotation, the expression matrix was ordered by PD-L1 value from high to low, all patients were divided into high and low groups. GO gene sets (C5) were selected as reference molecular signature databases to perform GSEA. ClusterProflier (R package (4.1.4)) was utilized to perform.

## Cell transfection, treatment, and lentiviral-based gene transduction

The MDA-MB-231, MDA-MB-468, HEK-293T, A549, HepG2, EO771, and MFC cell lines, were maintained in DMEM supplemented with 10% FBS. 4T1 cells were maintained in RMPI-1640 supplemented with 10% FBS. The AGS cell line, was maintained in F-12K supplemented with 10% FBS at 37 °C in 5% $CO_2$ (vol/vol). All cells were negatively tested for mycoplasma contamination and authenticated based on short tandem repeat fingerprinting before use.

Lentiviruses were produced in HEK-293T cells using the VSVG and psPAX2 package vectors. The virus was harvested 48 h and 72 h after transfection to transduce HEK-293T, A549, HepG2, 4T1, MFC, EO771, MDA-MB-231 or MDA-MB-468 cells, followed by selection with 2 μg ml⁻¹ puromycin. The cell lines with stable TMUB1/HUWE1 knockdown were identified by RT-qPCR and used in subsequent functional experiments.

## Membrane purification

Using a tissue grinder, the cancer cells were homogenized in buffer A (0.025 M Tris-HCl, pH 7.4, 0.025 M NaCl, and a protease inhibitor cocktail) before being centrifuged at 16,000 g for 15 min. The supernatants were centrifuged once more for 1 h at 10,000 g to obtain the cytosolic fractions. After being centrifuged again at 16,000 g for 30 min, the pellets were re-suspended in buffer B (buffer 1 with 0.25% [v/v] Tween-20), sonicated for 3 10 sec at 25 W in an ice-water slurry with 15 sec of cooling time in between, then centrifuged once more at that same pressure. The resulting supernatants represent the solubilized membrane fraction. All steps were performed at 4 °C.

## TUBE Pulldowns

$1 \times 10^7$ cells were treated with 10 mM MG132 for 6 h, and then lysed in 1 mL of lysis buffer (1x protease inhibitor cocktail, 50 mM Tris-HCl, pH 7.5, 0.15 M NaCl, 1 mM EDTA, 1% NP-40, 10% glycerol). After centrifuging the cell lysate at 20,000 g for 20 min, cell fragments and other insoluble proteins were taken out. The 20 mL TUBE2-agarose beads (Lifesensors, Cat #UM402) were incubated with the supernatant with end-over-end rotation at 4 °C for 2 h. The beads were then eluted with 50 mL of 2x SDS-PAGE Sample Buffer after being washed three times with 1 mL of lysis buffer. Immunoblotting was used to examine the input and eluate for endogenous ubiquitin.

## Cell fraction purification by continuous sucrose density gradient centrifugation

$5 \times 10^7$ MDA-MB-231 cells were harvested. The cells were placed in a 7 ml Dounce homogenizer together with 2 ml of homogenization buffer (20 mM HEPES pH 7.2, 250 mM sucrose, and 0.5 mM EGTA). Centrifugation at 4 °C was used to pellet intact cells and nuclei (5 min, 200 g). In order to separate the cell nuclear fractions, supernatant was placed over a gradient of 25–65% sucrose and centrifuged at 100,000 g for 1 h at 4 °C. Fractions were collected from the gradient's bottom and subjected to SDS-PAGE analysis and Western blotting. As indicators of various cellular fractions, LAMP1 (lysosome), TOM20 (mitochondrion), α-Tubulin (cytoplasm), and Calnexin (ER) were utilized.

## ER purification by discontinuous sucrose density gradient centrifugation

$5 \times 10^7$ MDA-MB-231 cells were collected. The cells were placed in a 7 ml Dounce homogenizer together with 2 ml of homogenization buffer (20 mM HEPES pH 7.2, 250 mM sucrose, and 0.5 mM EGTA). Centrifugation at 4 °C was used to pellet intact cells and nuclei (5 min, 200 g). cell suspension for 5 min at 1,400 g and 4 °C in a centrifuge. The pellets were resuspended in 2 ml of MTE buffer (270 mM D-mannitol, 10 mM Tris, 0.1 mM EDTA, pH 7.4) with new PMSF after the supernatant was removed. The cell suspension on ice should be sonicated three times for 10 sec each, with 10-second breaks in between. After centrifuging the supernatant at 1400 g for 10 min, 100 L of the supernatant were set aside as "Total protein." The second supernatant was ultracentrifuged at 152,000 g for 70 min while being placed over a sucrose solution that had already been set up (2 ml 2.0 M sucrose, 3 ml 1.5 M sucrose, and 1.3 M sucrose from bottom to top). 3.6 ml of ice-cold 1x MTE + PMSF buffer were added to the 0.4 ml volume of the visible band at the interface of the 1.3 M sucrose gradient layer. The combined solution underwent 12,600 g ultracentrifugation at 4 °C for 45 min. After that, the pellet was resuspended in 100 μl PBS as the "ER fraction" for WB analysis after the supernatant had been decanted and discarded.

## ER purification by differential centrifugation

$1 \times 10^7$ MDA-MB-231 cells were collected. PBS was discarded, while cells were pelleted. To enhance cell swelling, cells were resuspended in cold hypotonic extraction buffer (10 mM HEPES (pH 7.8) with 1 mM EGTA and 25 mM potassium chloride) and incubated at 4 °C for 20 min. The supernatant was discarded after the cells were centrifuged for 5 min at 600 g to collect the swollen cells. After adding an isotonic extraction

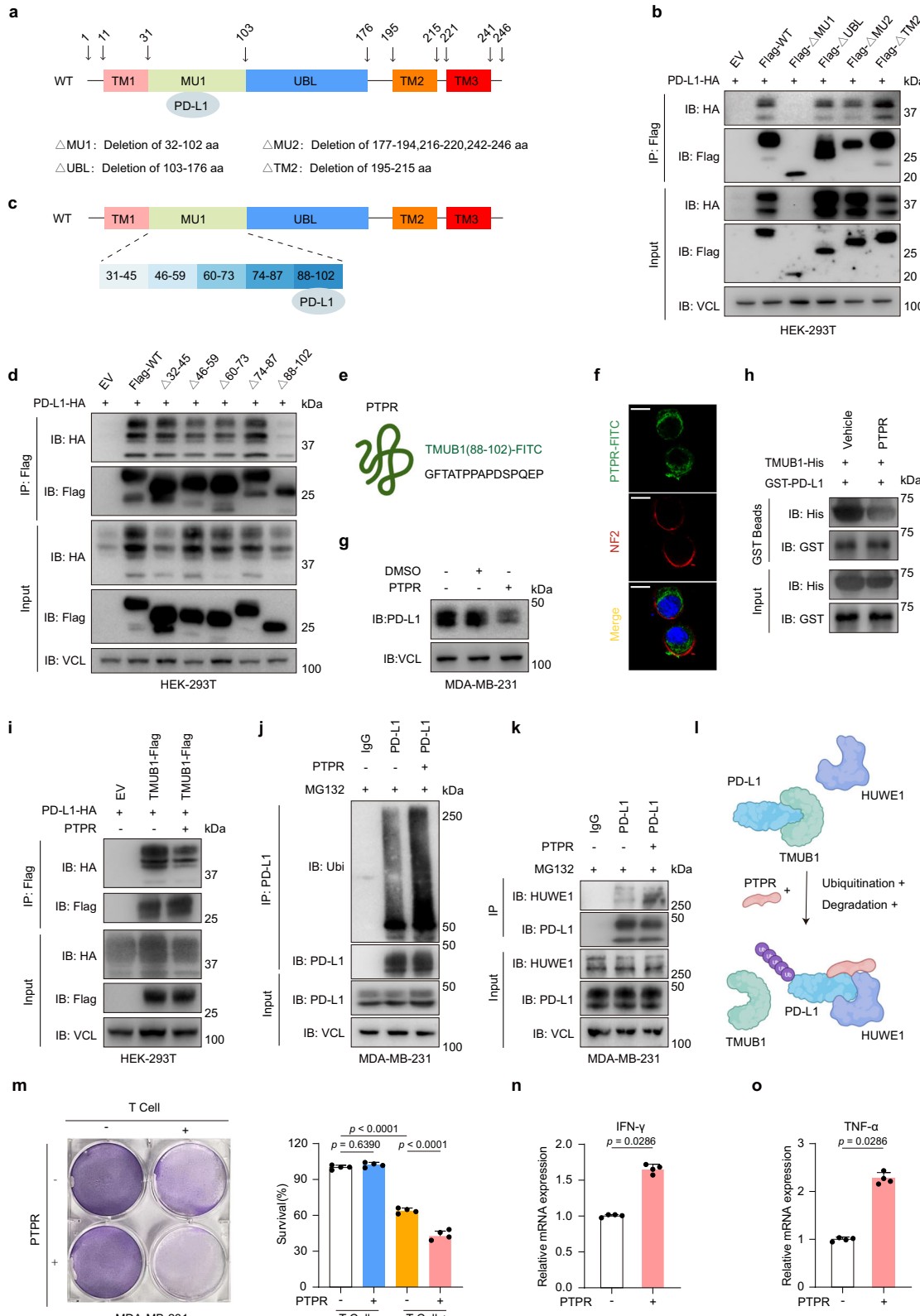

buffer (10 mM HEPES pH 7.8, 250 mM sucrose, 25 mM potassium chloride, and 1 mM EGTA) to the cells, they were homogenized with 10–20 Dounce homogenization strokes. The homogenate was centrifuged at 1,000 g for 10 min at a temperature of 4 °C, and the supernatant was transferred to a fresh tube. The supernatant was centrifuged at 12,000 g for 15 min at 4 °C, after which it was collected. This supernatant included microsomes and was almost devoid of LMF

components (the form of ER in vitro isolation processes). Estimating the amount of the appropriate supernatant, 7.5 mL 8 mM CaCl2 buffer was added. The combination solution was spun for 20–30 min at 4 °C until a flocculent precipitate, representing the ER microsome fraction, progressively developed. The supernatant was then centrifuged at 8,000 g for 10 min at 4 °C to isolate the ER microsomes, which were found in the pellet. To eliminate the cytosol contamination, the

**Fig. 6 | The PTPR competitive peptide prevents the upregulation of PD-L1 by TMUB1. a** Schematic diagram for TMUB1 displaying the positions of different domains. **b** Co-IP analysis for the interaction of different truncations of TMUB1-Flag with PD-L1-Myc in HEK-293T cells. **c** Schematic diagram for TMUB1 displaying the positions of different fragments of the MU1 domain. **d** Co-IP analysis for the interaction of the different truncations of TMUB1-Flag with PD-L1-HA in HEK-293T cells. **e** Schematic diagram for the PTPR peptide. **f** PTPR-FITC in MDA-MB-231 cells was observed using immunofluorescence staining, membrane protein NF2 was stained to characterize the cytomembrane. Scale bar: 10 μm. **g** Immunoblots of PD-L1 in the MDA-MB-231 cells untreated or treated with 10 μM of PTPR-FITC for 12 h. **h** Recombinant PD-L1-GST and TMUB1-His are purified for use in a GST pull-down assay. The interaction between TMUB1 and PD-L1 with or without the PTPR treatment was detected using the immunoblot assay. **i** Co-IP analysis for the interaction between TMUB1-Flag and PD-L1-HA in the HEK-293T cells untreated or treated with 10 μM of PTPR-FITC for 12 h. Co-IP analysis of the endogenous interaction between ubiquitin (**j**) or (**k**) HUWE1 and PD-L1 in MDA-MB-231 cells treated with 10 μM of MG132 for 6 h and 10 μM of PTPR for 12 hr or not. IgG was used as the negative control. **l** The working model of PTPR-mediated PD-L1 ubiquitination and degradation. **m** MDA-MB-231 cells were co-cultured with activated T cells for 48 h and subjected to treatment with 10 μM of PTPR for 12 h (or those left untreated), followed by crystal violet staining. The ratio of MDA-MB-231 cells to T cells: 1:3. Data are presented as mean ± SEM of n = 4. One-way ANOVA followed by Tukey's test. **n**, **o** The RT-qPCR analysis of the mRNA expressions of IFN-γ (**n**) or TNF-α (**o**) in PBMCs after co-culture with MDA-MB-231 cells. Data are presented as mean ± SEM of n = 4. Two-sided Mann–Whitney test.

supernatant was removed and the pellet was washed twice with an isotonic extraction buffer. The isolated ER was used for western blot analysis.

## Immunofluorescence

Cells were cultured in chamber slides overnight and fixed with 3.7% formaldehyde in PBS for 10 min at RT, followed by permeabilization with 0.5% Triton X-100 in PBS for 10 min. Cells were then blocked with 5% FBS in PBS for 30 min at RT and incubated with the indicated primary antibody for 1 h at RT, followed by incubation with anti-rabbit (or mouse) IgG (H + L), F(ab')2 fragment (Alexa Fluor 594 or 488 conjugate) from Abcam for 30 min at RT. Coverslips were mounted on slides using the anti-fade mounting medium with DAPI. IF images were acquired on an FV3000 confocal microscope (Olympus) or Super resolution Confocal Laser scanning microscope TCS SP8 STED (Leica). For each channel, all images were acquired with the same settings. Fluorescence images were obtained using FV31S-SW Viewer (v2.3.1), FV31S-DT (v2.3.1) (Olympus) and Leica Application Suite X (v3.3.0.16799) (Leica).

## Immunohistochemistry staining

The paraffin-embedded tissues were deparaffinized in xylene, rehydrated in a normal alcohol series, and then subjected to antigen retrieval by heating at 100 °C in citrate buffer for 15 min. The appropriate primary antibody was diluted in 3% BSA and applied to tissue slides, which were then incubated overnight at 4 °C. The slides were rinsed with PBS and treated for 60 min at room temperature with 3% BSA-diluted anti-rabbit or anti-mouse HRP-secondary antibody. The slides were dried in 50%, 70%, 80%, 95%, and 100% ethanol, and a mounting medium was used to stabilize them. Using an Olympus BX43 microscope and Olympus cell-Sens Dimension software, the pictures were captured. Using ImageJ (Fiji v1.51j) software, IHC staining density was evaluated and estimated based on the average staining intensity and the percentage of positively stained cells. Combining the proportion of positive cells and the intensity, a total score of protein abundance was determined.

## Mass spectrometry analysis

HEK-293T cells with stable PD-L1-Flag or Vector-Flag were harvested for mass spectrometry analysis. Cell lysate was prepared using polysome buffer (25 mM Tris-HCl (pH 7.5), 150 mM KCl, 0.5 mM DTT and 0.5% NP-40) with complete protease inhibitor cocktail. Flag-tag magnetic beads (Thermo Fisher Scientific) were prepared according to the manufacturer's instructions and then incubated with the cell lysate for 4 h at 4 °C with general rotation. Then, beads were washed with NT2 buffer three times and PBS once for 5 min at 4 °C. Afterwards, the bound protein was eluted using 3X Flag Peptide (APExBio). Finally, a 5× SDS loading buffer was added. The product was subjected to MS analysis.

The peptides were loaded on a trap column (75 μm i.d. × 2 cm, 3 μm, 100 Å, Thermo Fisher Scientific, USA), then separated on an analytical column (75 μm i.d. × 15 cm, 2 μm, 100 Å, Thermo Fisher Scientific, USA) operated in reverse-phase chromatography mode using the

UltiMate 3000 RSLC nano System (Thermo Fisher Scientific, USA). Mobile phase A consisted of 0.1% formic acid in water. Mobile phase B consisted of 0.1% formic acid in 80% acetonitrile solution. For solvent A, the linear gradient was increased from 2 to 10% over 10 min, 10–30% over 45 min, 30–98% over 5, and 6 min at 95% solvent B. The nano-LC was connected to an Orbitrap Fusion Lumos mass spectrometer (Thermo Fisher Scientific, USA) in positive-ion mode. The resolution of the acquired full scan was 30,000 with m/z from 150 to 1500 and the activation type was HCD with a collision energy of 30%.

The mass spectrum peptide sequences and protein identity were determined by matching the fragmentation patterns to protein databases using the Mascot software (Matrix Science, Boston, MA). Enzyme specificity was set to partially tryptic, with two missed cleavages. Spectral matches were filtered to false-discovery rate <1% at the peptide level using the target-decoy method.

The mass spectrometry proteomics data have been deposited to the ProteomeXchange Consortium via the PRIDE[55] partner repository with the dataset identifier PXD031702 (https://www.ebi.ac.uk/pride/archive/projects/PXD031702).

## Flow cytometry analysis of membrane PD-L1

For flow cytometric analysis for membrane PD-L1, MDA-MB-231, 4T1, MFC, or EO771 cells were collected by centrifugation at 1000 g for 5 min, incubated with PBS (0.5% bovine serum albumin) for 10 min at room temperature. The cells were probed with Fluor 488 - conjugated PD-L1 antibody and a matched isotype control at 4 °C for 30 min in the dark. After washing three times with PBS, the cells were analyzed using flow cytometry (Beckman Coulter Cytoflex) and data were analyzed using CytExpert V2.3 and FlowJo X software. The value of PD-L1-FITC median fluorescence (MFI) was used for evaluating the membrane abundance of PD-L1.

## Cell lysis, immunoprecipitation, and immunoblotting

In PBS, cells were extracted and homogenized in NETN buffer (25 mM Tris-HCl (pH 8.0), 100 mM NaCl, 1 mM EDTA, and 0.5 mM dithiothreitol (DTT)) with protease inhibitor cocktail, phosphatase inhibitor cocktail, Panobinostat, and methylstat. The lysates were purified by centrifuging them at 13,000 g for 15 min at 4 °C. The appropriate antibodies were used for IB or IP with the supernatants. Separate additions of the needed primary antibody and the control IgG were added to the prepared lysates for immunoprecipitation. After 3–6 h of incubation at 4 °C with gentle rotation, 20 μl of protein A/G magnetic beads (Pierce) were added to each lysate, followed by 2 h of incubation at 4 °C with moderate rotation. The protein-captured beads were washed with NETN buffer at 4 °C for 5 min each cycle. Then, 50 μl of 2× SDS loading buffer was used to elute the beads, and the eluted protein or protein complexes were identified by IB. Clarity Western ECL substrate was used to identify the blotting signals (Bio-Rad). As with tagged-protein IP, the main antibody and protein A/G beads (Pierce) were substituted by FLAG-M2 or HA magnetic beads (Sigma). Protein precipitated with Flag was eluted

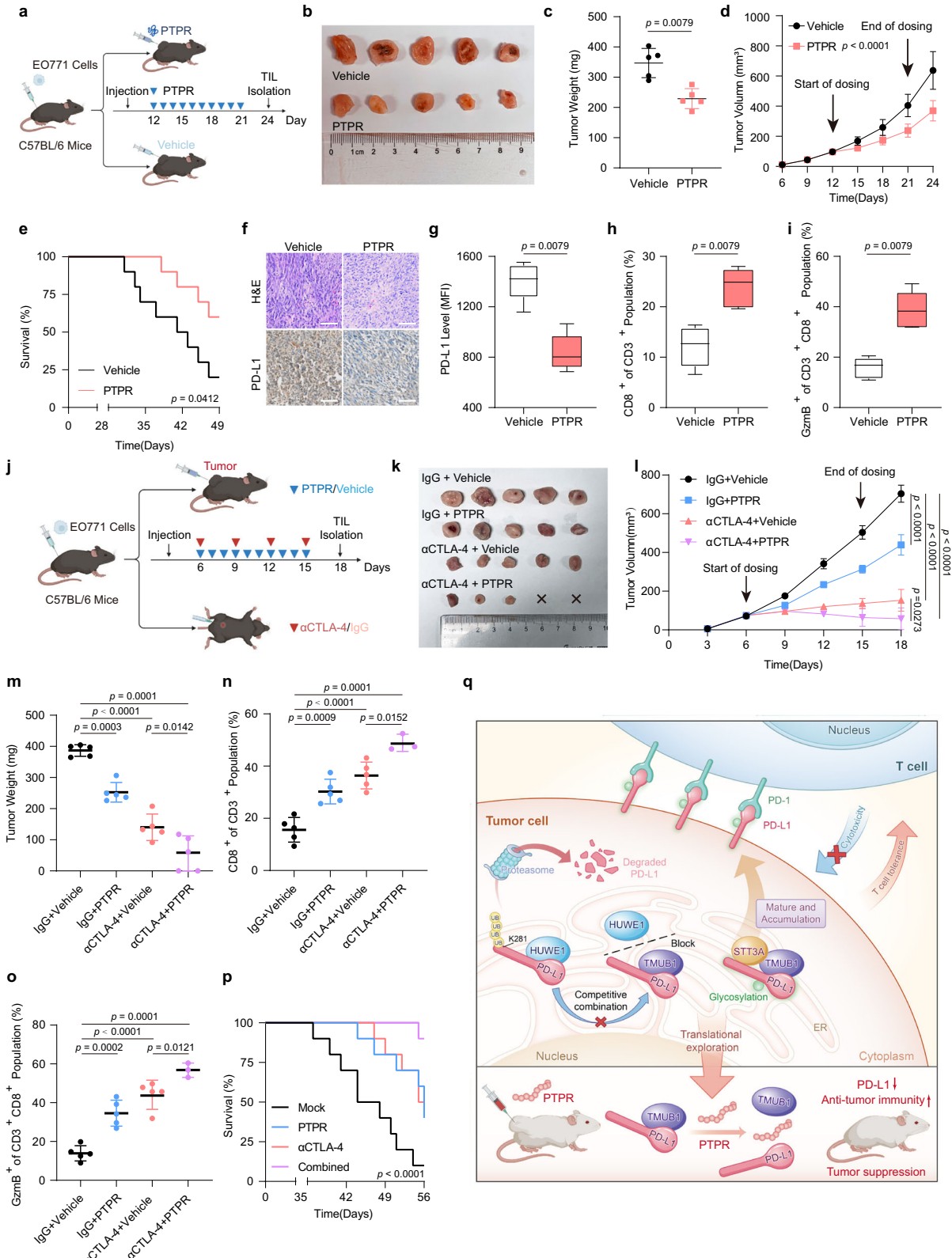

with 3× Flag Peptide (APExBio). Image Lab v4.1 software was used to acquire blot pictures (Bio-Rad). Blot images were obtained using Image Lab v4.1 software (Bio-Rad).

## Protein purification and In vitro protein pull-down assay
His-MBP-TMUB1 was purified using Ni-NTA Sefinose Resin (Sangon Biotech) after being expressed in *Escherichia coli* strain BL21-CodonPlus

(DE3)-RIPL (Agilent Technologies). GST-PD-L1 was isolated using GST magnetic beads (Sangon Biotech) after being produced in *Escherichia coli* strain BL21-CodonPlus (DE3)-RIPL (Agilent Technologies) (Sangon Biotech). HUWE1-Flag was isolated using Flag-M2 magnetic beads (Sigma) from the cell lysate of HEK-293T cells with HUWE1-Flag overexpression. The concentration and purity of recombinant proteins were determined using SDS-PAGE and Coomassie staining using BSA as a reference.

**Fig. 7 | In vivo antitumor effect and toxicity of PTPR. a** The injection schematic for PTPR and in vivo antitumor effect analysis. **b** Xenograft mouse model established using EO771 cells in C57BL/6 mice injected with PTPR or Vehicle (n = 5 mice per group). In vivo generated tumors are depicted. **c, d** Analysis of tumor weight (**c**) and growth (**d**) in the xenograft mouse model. Data are presented as mean ± SEM of n = 5 mice per group. Two-sided Mann-Whitney test for (**c**) and Two-way ANOVA for (**d**). **e** Survival in the mice injected with PTPR or vehicle. n = 10 mice per group. Log-rank test. **f** Representative IHC staining in randomly selected tumors from mice treated with PTPR or vehicle. Scale bar: 100 μm. **g–i** Flow cytometric analysis with median fluorescence intensity (MFI) of PD-L1 (**g**), CD3⁺ CD8⁺ T cells (**h**) or granzyme B-positive CD3⁺ CD8⁺ T cells (**i**). Data are presented as a box plot with box and whiskers. Bounds of box show the 25th and 75th percentiles, and the central lines in the box represent the median value. Whiskers show min to max value, n = 5 per

group. Two-sided Mann-Whitney test. **j** The injection schematic for PTPR and αCTLA-4 in vivo antitumor effect analysis**. k** Xenograft mouse model established using EO771 cells in C57BL/6 mice with indicated treatment (n = 5 mice per group). In vivo generated tumors are depicted. **l, m** Analysis of tumor growth (**l**) and weight (**m**) in the xenograft mouse model. Data are presented as mean ± SEM of n = 5 mice per group. Two-way ANOVA and One-way ANOVA followed by Tukey test. **n, o** Flow cytometric analysis of CD3⁺ CD8⁺ T cells (**n**) or granzyme B-positive CD3⁺ CD8⁺ T cells (**o**). Data are presented as a box plot with box and whiskers. Bounds of box show the 25th and 75th percentiles, and the central lines in the box represent the median value. Whiskers show min to max value, n = 5 per group. One-way ANOVA followed by Tukey's test. **p** Survival in the mice with indicated treatment. n = 10 mice per group. Log-rank test. **q** A schematic model of TMUB1 regulation of PD-L1 and HUWE1.

GST-tagged protein with the His-tagged protein or Flag-tagged protein (1–3 μg) wer incubated with GST magnetic beads (Sangon Biotech) in 500 μl of binding buffer (50 mM Tris-HCl at pH 7.9, 10% glycerol, 100 mM KCl, 5 mM MgCl2, 10 mM β-mercaptoethanol and 0.1% NP-40) for 2 h at 4 °C with gentle rotation. The beads were then washed with NETN buffer 3 times for 5 min each time at 4 °C with rotation. Subsequently, the beads were eluted in 50 μl 2× SDS loading buffer, and the eluted protein complexes were detected by IB.

### Xenograft mouse model
Prepared tumor cells in 30 μl of sterile PBS were injected separately into the flanks of 4- to 6-week-old female Balb/c, Balb/c nude or C57BL/6 J mice, using the 100 μl sterile syringe. The tumor size was measured every 2 or 3 days using a c alliper, and tumor volume was calculated using the standard formula: $0.54 \times L \times W^2$, where L is the length (longest diameter) and W is the width (shortest diameter). The mice were euthanized by cervical dislocation when they met the institutional euthanasia criteria for tumor size (L or W > 20 mm) or overall health condition. The tumors were then removed, photographed, and weighed.

### qRT-PCR assay
TRIzol reagent (Invitrogen) was used to extract the associated RNAs according to the manufacturer's instructions. Reverse transcription was performed using the iScript cDNA synthesis kit (Bio-Rad), and the abundance of target RNAs was detegroupy the iTaqTM Universal SYBR Green Supermix qPCR kit (Bio-Rad) according to the manufacturer's instructions. Relative quantities of gene expression levels in each group were normalized to the reference genes (Actin-β, GAPDH or U6) and then normalized to the control group, as $Exp = 2^{-((CT(target,test)-CT(ref,test))-(CT(target,control)-CT(ref,control)))}$. Primer information was shown in Supplementary Table 5.

### T Cell killing assay
Human peripheral blood mononuclear cells (PBMC; STEMCELL Technologies) were grown in RMPI-1640 medium with 10% FBS, Immuno-Cult Human CD3/CD28/CD2 T cell activator (10970; STEMCELL Technologies), and IL-2 (10 ng/mL; PeproTech) for one week to acquire activated T cells. Experiments were conducted in DMEM medium with 100 ng/mL anti-CD3 antibody and 10 ng/mL IL-2. cancer cells were allowed to cling to the plates overnight, which were subsequently treated with activated T cells for 48 h. The ratios (1:3 or 1:5) between cancer cells and activated cells were changed for each experiment's objective (see Figure Legends). T cells and cell debris were removed using a PBS wash, and then the number of viable cancer cells was determined using crystal violet staining.

### TIL isolation
Pieces of tumor tissue were suspended in 5 ml tumor digestion buffer (5% FBS, 20 mM glutamine, 50 M β-mercaptoethanol, 1.6 mgml1 collagenase IV, 1.6 mgml1 collagenase I, and 0.02% DNase I). After 1.5 h of rotation at 37 °C, the tissues were digested. Using a 70-μm filter, a single-

cell suspension was obtained from the cell suspension. After washing the cells by blowing with PBS, centrifuge at 1100 g for 7 min and carefully and slowly transfer the supernatant to a pre-prepared 40%/70% Percoll (GE) separator. By centrifugation at the lowest lift speed of 750 g for 22 min, the isolated lymphocytes can be observed at the boundary as a milky white band. After PBS washing, the tumor-infiltrating leukocytes were then stained with fluorescently labeled antibodies against several markers, including CD8, CD3, and GzmB.

### Statistics and reproducibility
All statistical results are reported as the mean ± SEM of three or more independent biological replicates. Representative images for fluorescence staining, IHC staining, and immunoblot are shown. Each of these experiments was independently repeated three times. Relative quantities of gene expression levels were normalized to β-Actin, GAPDH, or U6. For every figure, statistical tests are justified as appropriate. Analyses and graphical presentation were performed using the GraphPad Prism 8.0 software. The experiments were not randomized. The investigators were not blinded to allocation during experiments and outcome assessment.

### Reporting summary
Further information on research design is available in the Nature Research Reporting Summary linked to this article.

## Data availability
The mass spectrometry proteomics data generated in this study have been deposited in the ProteomeXchange Consortium via the PRIDE partner repository with the dataset identifier PXD031702 (https://www.ebi.ac.uk/pride/archive/projects/PXD031702). All data are included in the Supplemental Information or available from the authors upon reasonable requests, as are unique reagents used in this Article. Source data are provided with this paper.

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

## Acknowledgements

We thank Z.-G. Shao (Beijing University) for gifting HUWE1 template vectors. We also thank Guangzhou Sagene Biotech Co., Ltd. for their help with pattern diagram making. This work was supported in part by the National Key Research and Development Program of China (No. 2021YFC2700903, No. 2021YFF1200404), the National Natural Science Foundation of China (No. 81672791, No. 81872300), the Zhejiang Provincial Natural Science Fund for Distinguished Young Scholars of China (No. LR18C060002), Huadong MedicineJoint Funds of the Zhejiang Provincial Natural Science Foundation of China (No. LHDMY22H160006), the Starry Night Science Fund of Zhejiang University Shanghai Institute for Advanced Study (No. SN-ZJU-SIAS-003), ZJU-QILU Joint Research Institute and Qilu Group.

## Author contributions

A.L., T.Z., W.W., X.L. and R.Z. conceived and designed the research. C.S., Yin.W. and M.W. performed most of the biochemical, molecular experiments, with assistance from Y.C., F.L., Yir. W., Y.S., L.S., Z.Z., Z.G., L.Y. and X.H. L.S. and S.X. performed mass spectrum analysis. H.J. and L.W. provided the clinical samples of gastric cancer and breast cancer. C.S, Yin.W. and Y.C performed ascertainment and processing of clinical specimens. L.Q. and Z.Y. performed bioinformatics analysis. C.S, Z.S., Y.G., C.P., J.C. and X.D. performed PTPR design. C.S., Yin.W., Y.C., F.L., Yir.W., and Y.S. performed xenograft experiments, IHC analyses, PTPR treatment and TIL isolation and analyses. A.L., T.Z., R.Z., W.W., X.L., J.S., P.X., L.Y., Q.Y., Z.C. and J.L. contributed to the data discussion. A.L., T.Z., R.Z.,W.W and X.L. initiated and supervised the project. C.S., Yin.W. and A.L. wrote the manuscript. Aifu Lin is the lead contact for this paper.

## Competing interests

The authors declare no competing interests.

## Additional information

[1]MOE Laboratory of Biosystem Homeostasis and Protection, College of Life Sciences, Zhejiang University, Hangzhou, Zhejiang 310058, China. [2]Cancer Center, Zhejiang University, Hangzhou, Zhejiang 310058, China. [3]Key Laboratory for Cell and Gene Engineering of Zhejiang Province, Hangzhou, Zhejiang 310058, China. [4]Innovation Institute for Artificial Intelligence in Medicine, Zhejiang University, Hangzhou, Zhejiang 310016, China. [5]Hangzhou Institute of Innovative Medicine, Institute of Drug Discovery and Design, College of Pharmaceutical Sciences, Zhejiang University, Hangzhou, Zhejiang 310058, China. [6]Key Laboratory of Structural Biology of Zhejiang Province, Westlake Laboratory of Life Sciences and Biomedicine, Westlake University, Hangzhou, Zhejiang 310024, China. [7]Institute of Immunology, Zhejiang University School of Medicine, Hangzhou, Zhejiang 310009, China. [8]Sun Yat-sen University Cancer Center, State Key Laboratory of Oncology in South China, Collaborative Innovation Center for Cancer Medicine, Guangzhou, Guangdong 510060, China. [9]Zhejiang University-University of Edinburgh Institute (ZJU-UoE Institute), Zhejiang University School of Medicine, International Campus, Zhejiang University, Haining, Zhejiang 314400, China. [10]Department of Gastroenterology, the Second Affiliated Hospital, School of Medicine and Institute of Gastroenterology, Zhejiang University, Hangzhou, Zhejiang, China. [11]MOE Laboratory of Biosystems Homeostasis & Protection and Zhejiang Provincial Key Laboratory for Cancer Molecular Cell Biology, Life Sciences Institute, Zhejiang University, Hangzhou, Zhejiang 310058, China. [12]Department of Developmental and Cell Biology, University of California, Irvine; Irvine, CA 92697, USA. [13]Shanghai Institute for Advanced Study, Zhejiang University, 201203 Shanghai, China. [14]Department of Chemistry, Colombia University, New York City, NY 10027, USA. [15]Institute of Quantitative Biology, Zhejiang University, Hangzhou, Zhejiang 310058, China. [16]Department of Cell Biology and Program in Molecular Cell Biology, Zhejiang University School of Medicine, Hangzhou, Zhejiang 310058, China. [17]Department of Gastroenterology, the Second Affiliated Hospital, School of Medicine and Institute of Gastroenterology, Zhejiang University, Hangzhou, Zhejiang 310009, China. [18]Breast Center of the First Affiliated Hospital, School of Medicine, Zhejiang University, Hangzhou, Zhejiang 310003, China. [19]International School of Medicine, International Institutes of Medicine, The 4th Affiliated Hospital of Zhejiang University School of Medicine, Yiwu, Zhejiang 322000, China. [20]ZJU-QILU Joint Research Institute, Hangzhou, Zhejiang 310058, China. [21]These authors contributed equally: Chengyu Shi, Ying Wang, Minjie Wu. ✉e-mail: lixu@westlake.edu.cn; wenqiw6@uci.edu; rhzhou@zju.edu.cn; tzhou@zju.edu.cn; linaifu@zju.edu.cn

