## [Peer Review File · Nature Communications]

Promoting anti-tumor immunity by targeting TMUB1, a modulator of PD-L1 polyubiquitination and glycosylationREVIEWER COMMENTS

Reviewer #1 (expertise in PD-L1, PD-L1 pharmacological inhibition):

This manuscript from Shi, et al., test effects of reducing tumor PDL1 content by manipulating TMUB1. It is timely to consider alternatives to antibody-mediated PDL1 blockade as an approach to improving cancer immunotherapy. This report is novel, interesting and highly mechanistic on the PDL1 regulation side. It is not as mechanistic on the cell-intrinsic PDL1 side or for immune outcomes, but overall it offers many mechanistic insights.

Interpretations are generally well-supported by data shown with minor caveats as shown below.

The writing and data presentations are clear.

Fig. 1 Data that tumor TUBM1 improves PDL1 content, is associated with reduced CD8+ TIL and patient survival is all good. However, there are some unclear aspects to these data. It is unclear what the GSEA analysis for panel A is. In panels B-C, what are values on Y-axis? Data that genetic TUBM1 manipulations alter tumor PDL1 are good. Data that TUBM1 suppresses T cell functions in vitro is good. IP shows there could be a TUBM1/PDL1 interaction within cells.

Fig. 2 PDL1/HUWE1/TMUB1 interaction data are strong and well-controlled

Fig. 3 PDL1/HUWE1/TMUB1 localization and post-translational control data are strong and well-controlled

Fig. 4 Data that TMUB1 suppresses anti-tumor immunity is strong. However, whether TMUB1 immune effects are from PDL1 control cannot be ascertained from these data. Genetic alteration of TMUB1 in genetically PD-L1-depleted cells would address this issue directly, which is addressed in Fig. 7, which makes the case better, that effects are PDL1-driven. The CXCL10 and CCL5 increase are reminiscent of STING activation which could occur from reduced DNA damage repair from reduced PDL1 as recently reported (Cancer Res. 2022 Mar 3. pii: 682005. doi: 10.1158/0008-5472.CAN-21-2076). Can the authors comment?

Fig. 5 Human relevance of TMUB1 control of PDL1 is strong.

Fig. 6 Epitope mapping of PDL1 binding domain of TMUB1 and functional relevance are strong

Fig. 7 These data are more convincing for TMUB1 effects through PDL1 control and are very strong. Inclusion of aCTLA4 effects further extends scope of relevance and translational potential.

The discussion is generally relevant and comprehensive. A curious omission is any reference to tumor cell-intrinsic PDL1 effects, which could well be operative here, in addition to the effects of loss of surface expressed PDL1. It would have been interesting to understand if TMUB1 regulates PDL1 equally on the surface versus cytosol versus nucleus where distinct PDL1 signals from these distinct subcellular locations have been reported. Nonetheless, the data shown are highly detailed enough that such work could be omitted here, but it should at least be discussed, as should potential for cell-intrinsic PDL1 effects, such as for example, the chemokine effects shown here and reported to be a cell-intrinsic PDL1 effect (e.g., PMC5321797), among other considerations.

Reviewer #2 (expertise in PD-L1 ubiquitination and signalling, cancer):

This study by Chengyu Shi et al. identified Transmembrane and ubiquitin-like domain containing protein 1 (TMUB1) as a novel modulator for post-translational modifications of PD-L1 in cancer. They showed that TMUB1 stabilizes PD-L1 by antagonizing HUWE1, a novel E3 ubiquitin ligase that interacts with PD-L1 and increases its polyubiquitination. Furthermore, TMUB1 enhances PD-L1 stability by increasing PD-L1 N-glycosylation via recruiting STT3A, ultimately promoting PD-L1

maturation. Higher TMUB1 levels were associated with higher PD-L1 levels and worse clinical outcomes in patients, suggesting that the increase in the TMUB1 expression in cancer results in PD-L1 accumulation, which drives the escape of tumor cells from antitumor immunity. Targeting TMUB1 reverses the suppressive tumor immune microenvironment. Authors developed a TMUB1 competitive peptide, PTPR, to inhibit the binding of PD-L1 to TMUB1. PTPR enhances the antitumor immunity and suppresses tumor growth in mice, suggesting its role as a promising immunotherapeutic target.

The results support the author's conclusions and the manuscript is written in a clear and logical way. This manuscript has a lot of potential, and fits the scope of this journal well. I have listed some concerns below.

1. The authors wrote "TMUB1 is reported to be involved in the ubiquitin-proteasome degradation of several proteins", referring the paper of Della-Fazia, M.A., et al. 2021 (The Ins and Outs of HOPS/TMUB1 in biology and pathology. The FEBS journal 288, 2773-2783). On the contrary, this paper showed that TMUB1 stabilizes p19ARF, and thus inhibits p53 proteasomal degradation, and induces cell proliferation arrest. The authors need to correct their explanation. Authors also wrote "Recent studies have demonstrated that TMUB1 could modulate the stability of the E3 ligase TRAF6 through direct binding⁵. The reference 5 is not matching with the paper they referred, maybe it is ref. 20.

Does TMUB1 antagonize TRAF6 as well to increase PDL1 level? Since TMUB1 does not bind to HUWE1, is there an interaction between TRAF6 and HUWE1 in this TMUB1-HUWE1 pathway controlling PD-L1?

PD-L1 ubiquitination and degradation by E3 ligase/s have been reported previously. Authors may want to cite a recent paper showing that PD-L1 is ubiquitinated and degraded by the E3 ligase FBXO22. (De S, Holvey-Bates EG, Mahen K, Willard B, Stark GR. The ubiquitin E3 ligase FBXO22 degrades PD-L1 and sensitizes cancer cells to DNA damage. PNAS 2021, 23;118(47):e2112674118).

2. Mezzadra et al. (Identification of CMTM6 and CMTM4 as PD-L1 protein regulators. Nature, 549(7670), 106–110. 2017), ref 37, reported that the cellular membrane protein CMTM4/6 interacts with PD-L1, leading to inhibition of PD-L1 ubiquitination and degradation, which consequently impaired T cell activity. Is there any cross-talk between TMUB1 and CMTM6 or CMTM4 in the process of inhibition of PD-L1 degradation?

3. The authors showed that TMUB1 could bind directly to the glycosylation mutant PD-L1 (4NQ) and reduce the degradation of mature PD-L1 as well as that of non-glycosylated PD-L1. They also showed that HUWE1 binds to glycosylated and non-glycosylated forms of PD-L1 upon MG132 treatment. Does HUWE1 bind to 4NQ mutant PD-L1? Can over expression of HUWE1 in 4NQ mutant cells increase PDL1 degradation and antagonize TMUB1 mediated inhibition of the degradation of PD-L1?

4. The peptide PTPR decreases PD-L1 expression. The authors may need to check whether PTPR increases the expression level of HUWE1 to facilitate the degradation of PD-L1.

5. Figure 3 a and b results were shown in HEK293T cells, not in MDAMB231 cells. Needs to be fixed in the legend.

Reviewer #3 (expertise in HOPS/TMUB1, inflammation):

Please find enclosed the comments of the paper entitled: Break Antitumor Immunosuppression by Targeting TMUB1, a Novel Modulator for Posttranslational Modifications of PD-L1, by Shi C, -Aifu Lin et al., that has been submitted for publication in Nature Comm.

The research shows a novel target, PD-L1, of TMUB1 modifier in a context of tumor cells.

Indeed, the manuscript is difficult to read and understand. A huge series of experiments are

present, but a lot of them are not useful and create confusion. The controls of a lot of experiments are not present and the quality of the WB and the images of IF are not well performed. To define a role of a protein is not possible to perform almost all experiments by transfection and analyze by tag the expression. These ways to perform experiments alter completely the biology and physiology of the cells.

The aim of the manuscript is loss in a lot of streams, but does not go straight to the question. The result of the ability of TMUB to control PD-L1 stability is present, but the mechanism is far to be explained. Also, the potential use of the peptide is not convincing the idea about the HUWE is not well determined.

Technical Analysis

The technical analyses of the manuscript showed a high number of flaws, almost all experiments lack of controls and the majority of the experiments have been performed in transfection. Even the experiment to test the half-life of PD-L1 in presence of TMUB1 has been performed in transfection. This is not corrected way to establish protein stability. In almost all of the WB are not present Molecular Weight Markers in supplementary data. Arbitrarily, they displayed different bands of TMUB1 and PD-L1 in relation to the experiments performed. The survival test of T-cells has been defined, but the why the cells died is not described, apoptosis? No experiments are reported In some IF experiments assessing intracellular localization, the merged images of TMUB and PDL1 or HUWE1 are not reported (Fig3C, D and Ext data Fig2A). The merge with compartmental marker of each of the partner one by one are not enough to assess their interaction, they MUST be revealed in the same specimen. The compartmental marker HAS NOT TO BE TRANSFECTED OR TAGGED for detection in IF (Fig3C, D). Why Calnexin-GFP has be used? Moreover, the GFP creates an ER stress of the cells. When analysing the internalization of PTPR a plasma membrane marker is required to verify the real intracellular localization (Fig 6f).

Even if when transfected the IF detection does not interest TMUB, a control assessing TMUB amount is mandatory (WB or qPCR as preferred) (I.e., Ext data Fig2, 3). This is required for all proteins modulated but not detected in IF assays.

In some immunofluorescent assay scale bars are not reported or, if present, not in all the squares of the images (i.e., Fig 6f, Ext data Fig2A or Ext data Fig3C, D). They must be the same in all of them. Pay attention and choose one format only.

I know that captions have to be short, but if too short they are no useful at all.

Conceptual Analysis:

The data that TMUB acts on PDL1 stability seems certain, but the mechanistic explanation of TMUB activity is quite confused and unclear. Many inhibitors or other factors used in the experimental procedure are only cited, sometimes without introducing their function or the reason for use.

“TMUB1 competes with HUWE1, a novel E3 ubiquitin ligase of PD-L1, to interact with PD-L1 and attenuates its polyubiquitination ”

If assessing the competition between TMUB and HUWE in PDL1 site of binding, the effect must be observed in both TMUB vs HUWE and HUWE vs TMUB direction...in this paper only how TMUB hinders HUWE binding is observed, but not the reverse reaction. Moreover, TMUB1, as modifier, to be linked to PD-L1 needs an E3, and the authors hypothesize HUWE only such a sort of competitor. Indeed, the author cites a manuscript, where HOPS/TMUB1 regulates TRAF6, but in this case HOPS/TMUB1 does not act as a competitor, but such as a regulator of NF- κ B. No discussion or scenario are suggested by the author about a possible mechanism of the PD-L1-TMUB1 interaction as a modifier.

“Mutation analysis revealed that K178 and K281 were associated with PD-L1 protein stability and ubiquitination, while only K281, but not K75, K178, K185, or K280 (Extended Data Fig. 3g), is required for the HUWE1-induced ubiquitination and degradation of PD-L1 (Extended Data Fig.3h-j” Several errors are present in this sentence. 1) Any K at 280 position is reported; 2) of the two lysines the K178 rather than the K281 is involved in degradation. If K281 was the effective one, its mutation should make TMUB ineffective and deplete its function... on the contrary PDL1 ubiquitination is still present in the absence of K281. Vice versa and for the same reason K178, whose mutation deplete PDL1 degradation, is the functional one.

The TUBE assays used to perform ubiquitination analysis deserves quite more details.

Only Fig3N gives a clear idea of an effect of HOPS in affecting the balance between glycosylated and not glycosylated forms of PDL1. The TMUB relationship with STT3A is assessed by the Co-IP binding evaluated, as nearly always in this paper, using transfected and tagged proteins and bioinformatics. In the paper isn't reported any experiment suggesting a clear interaction TMUB-STT3A in controlling PDL1 glycosylation (i.e., glycosylation rate STT3A-dependent in the lack or overexpression of TMUB). Moreover (pag.17 lines 328), the authors wrote Together, TMUB1 regulated the glycosylation of PD-L1 by recruiting STT3A.... What means recruiting a direct binding a complex?

Pag. 18 The experiment in mice is not well described, the reader is misled by the description. The cell lines and the mice used are not well explicated.

"the peptide capable of inhibiting the binding of TMUB1 to PD-L1 were designed"

The PTPR inhibiting peptide developed through the analysis of binding sites, is produced to counteracts the inhibiting effect of TMUB on HUWE mediate degradation. Even if in vivo xenografts experiments give encouraging results and suggest an antitumoral effect in mice, the mechanism is not clearly explained.

"Indeed, the addition of PTPR weakened the binding between PD-L1 and TMUB1 and reduced the level of PD-L1 protein caused by TMUB1 overexpression (Fig. 6h and Extended Data Fig. 9e)"

"PTPR was a TMUB1-targeting molecule that effectively inhibits the interaction between PD-L1 and TMUB1 and downregulates the cellular abundance of PD-L1 (Fig.6k)."

The authors wrote the opposite in the first part...

The peptide binds to the shared PDL 1 binding site. If so, it would inhibit HUWE binding and further degradation thus enhancing half-life. Why are these data not included? On the contrary the author showed an enhanced ubiquitination rate upon PTPR addiction in cell cultures.

Why if binding site is inhibited by PTPR the author declares that PDL1 levels are reduced by increased degradation?

This question deserves deeper analysis and is not answered in the submitted paper.

REVIEWER COMMENTS

Reviewer #1 (expertise in PD-L1, PD-L1 pharmacological inhibition):

This manuscript from Shi, et al., test effects of reducing tumor PDL1 content by manipulating TMUB1. It is timely to consider alternatives to antibody-mediated PDL1 blockade as an approach to improving cancer immunotherapy. This report is novel, interesting and highly mechanistic on the PDL1 regulation side. It is not as mechanistic on the cell-intrinsic PDL1 side or for immune outcomes, but overall it offers many mechanistic insights.

Interpretations are generally well-supported by data shown with minor caveats as shown below.

The writing and data presentations are clear.

Fig. 1 Data that tumor TUBM1 improves PDL1 content, is associated with reduced CD8+ TIL and patient survival is all good. However, there are some unclear aspects to these data. It is unclear what the GSEA analysis for panel A is. In panels B-C, what is values on Y-axis? Data that genetic TUBM1 manipulations alter tumor PDL1 are good. Data that TUBM1 suppresses T cell functions in vitro is good. IP shows there could be a TUBM1/PDL1 interaction within cells.

Response:

Sorry for any confusion.

For Panel A, we divided the PD-L1 expression data of breast invasive carcinoma (BRCA) from the TCGA database into two groups, compared the enriched pathways, and finally found that some biological process pathways associated with posttranslational modifications were enriched. X axis was the enrichment score, and Y axis was enriched pathways ranked by enriched score.

The values on the Y-axis in the Panel B and C refer to the abundance of N-glycosylated PD-L1 and phosphorylated PD-L1 detected by quantitative mass-spectrometry-based N-glycoproteomic and phosphoproteomic analyses in cancer and paracancer samples from multiple breast cancer patients in another group's study (PMID: 32543193, PMID: 27251275).

We've gone into more detail in Legends so that our content can be more clearly understood.

Fig. 2 PDL1/HUWE1/TMUB1 interaction data are strong and well-controlled

Response:

Thank you!

Fig. 3 PDL1/HUWE1/TMUB1 localization and post-translational control data are strong and well-controlled

Response:

Thank you!

Fig. 4 Data that TMUB1 suppresses anti-tumor immunity is strong. However, whether TMUB1 immune effects are from PDL1 control cannot be ascertained from these data. Genetic alteration of TMUB1 in genetically PD-L1-depleted cells would address this issue directly, which is addressed in Fig. 7, which makes the case better, that effects are PDL1-driven. The CXCL10 and CCL5 increase are reminiscent of STING activation which could occur from reduced DNA damage repair from reduced PDL1 as recently reported (Cancer Res. 2022 Mar 3. pii: 682005. doi: 10.1158/0008-5472.CAN-21-2076). Can the authors comment?

Response:

We developed Pd-11-KO 4T1 and PD-L1-KO MDA-MB-231 cell lines. On the one hand, reduction of PD-L1 followed by overexpression of TMUB1 failed to strengthen the resistance of tumor cells to T cell death (**Response Figure.1a, 1b**). On the other hand, PTPR therapy failed to achieve effective anti-tumor effectiveness *in vivo* in Pd-11-KO tumor cell lines (**Response Figure.1c and 1d**). All of this shows that the immunological control of TMUB1 is dependent on PD-L1 for its function.

Anand V R Kornepati et al. discovered that in addition to being a surface-binding molecule, PD-L1 plays a crucial role in DNA damage repair. PARP inhibitor-induced activation of cell-intrinsic TANK-binding kinase 1 (TBK1) and production of the T-cell chemoattractant CCL5 was inhibited by tumor PD-L1. This is consistent with our discovery that targeting TMUB1 led to a decrease in PD-L1 abundance and, consequently, an increase in CCL5 and CXCL10 expression, demonstrating the breadth of TMUB1-PD-L1 regulation. This section has been added to the **Discussion**.

Response Figure. 1

(a) The WT or PD-L1-KO MDA-MB-231 cells with EV or TMUB1-Overexpression co-cultured with activated T cells for 48 hr and then subjected to crystal violet staining. (b) The ratio of MDA-MB-231 cells to T cells: 1:3. Data are presented as mean ± SEM; n = 3. One-way ANOVA followed by Tukey test, **P < 0.01; n.s = no significance.

(b) Xenograft mouse model established using PD-L1-KO 4T1 cells in Balb/c mice injected with PTPR or Vehicle (n = 5 mice per group). *In vivo* generated tumors ac

(c) Analysis of tumor growth and volume in the xenograft mouse model. Data are presented as mean ± SEM. of n = 5 mice per group. Two-way ANOVA, n.s = no significance.

Fig. 5 Human relevance of TMUB1 control of PDL1 is strong.

Response:

Thank you!

Fig. 6 Epitope mapping of PDL1 binding domain of TMUB1 and functional relevance are strong

Response:

Thank you!

Fig. 7 These data are more convincing for TMUB1 effects through PDL1 control and are very strong. Inclusion of aCTLA4 effects further extends scope of relevance and translational potential.

Response:

Thank you!

The discussion is generally relevant and comprehensive. A curious omission is any reference to tumor cell-intrinsic PDL1 effects, which could well be operative here, in addition to the effects of loss of surface expressed PDL1. It would have been interesting to understand if TMUB1 regulates PDL1 equally on the surface versus cytosol versus nucleus where distinct PDL1 signals from these distinct subcellular locations have been reported. Nonetheless, the data shown are highly detailed enough that such work could be omitted here, but it should at least be discussed, as should potential for cell-intrinsic PDL1 effects, such as for example, the chemokine effects shown here and reported

to be a cell-intrinsic PDL1 effect (e.g., PMC5321797), among other considerations.

Response:

Thanks for your helpful suggestion! The function of PD-L1 as an intracellular protein in addition to the cell membrane has received extensive attention, including transcriptional regulation in the nucleus and the regulation of molecules involved in key nodes of the signaling pathway (**PMID: 35247877, 32839551, 32929201, 28220772**). This suggests that immunotherapy research should place a greater emphasis on the molecular mechanisms of total cellular PD-L1 regulation in addition to PD-L1 in the cell membrane. Our discovery that TMUB1 is involved in the degradation and glycosylation of PD-L1 on the endoplasmic reticulum, which in turn regulates the total PD-L1 abundance, suggests that targeting TMUB1 should result in inhibition of total PD-L1 abundance and therefore limit PD-L1 regulatory functions in multiple cellular activities. All of these findings indicate that TMUB1 has the potential to be a dependable and successful immunotherapy target. This section has been added to the **Discussion**.

We thank the Reviewer for recognizing our research and requesting several crucial and specific experiments and comments that have enabled more rigorous, qualitative conclusions in support of the proposed model, and which will unquestionably increase the confidence in the conclusions that can be drawn.

Reviewer #2 (expertise in PD-L1 ubiquitination and signalling, cancer):

This study by Chengyu Shi et al. identified Transmembrane and ubiquitin-like domain containing protein 1 (TMUB1) as a novel modulator for post-translational modifications of PD-L1 in cancer. They showed that TMUB1 stabilizes PD-L1 by antagonizing HUWE1, a novel E3 ubiquitin ligase that interacts with PD-L1 and increases its polyubiquitination. Furthermore, TMUB1 enhances PD-L1 stability by increasing PD-L1 N-glycosylation via recruiting STT3A, ultimately promoting PD-L1 maturation. Higher TMUB1 levels were associated with higher PD-L1 levels and worse clinical outcomes in patients, suggesting that the increase in the TMUB1 expression in cancer results in PD-L1 accumulation, which drives the escape of tumor cells from antitumor immunity. Targeting TMUB1 reverses the suppressive tumor immune microenvironment. Authors developed a TMUB1 competitive peptide, PTPR, to inhibit the binding of PD-L1 to TMUB1. PTPR enhances the antitumor immunity and suppresses tumor growth in mice, suggesting its role as a promising immunotherapeutic target.

The results support the author's conclusions and the manuscript is written in a clear and logical way. This manuscript has a lot of potential, and fits the scope of this journal well. I have listed some concerns below.

1. The authors wrote "TMUB1 is reported to be involved in the ubiquitin-proteasome degradation of several proteins", referring the paper of Della-Fazia, M.A., et al. 2021 (The Ins and Outs of HOPS/TMUB1 in biology and pathology. The FEBS journal 288, 2773-2783). On the contrary, this paper showed that TMUB1 stabilizes p19ARF, and thus inhibits p53 proteasomal degradation, and induces cell proliferation arrest. The authors need to correct their explanation. Authors also wrote "Recent studies have demonstrated that TMUB1 could modulate the stability of the E3 ligase TRAF6 through direct binding⁵. The reference 5 is not matching with the paper they referred, maybe it is ref. 20.

Response:

Sorry for any confusion. We have changed our formulation and added appropriate citations. We have also corrected any errors in the citation and thank you for pointing them out.

Does TMUB1 antagonize TRAF6 as well to increase PDL1 level? Since TMUB1 does not bind to HUWE1, is there an interaction between TRAF6 and HUWE1 in this TMUB1-HUWE1 pathway controlling PD-L1?

Response:

Thanks for your helpful suggestion! The TRAF6-mediated activation of NF- κ B signaling pathway has been widely reported (PMID: 29894691, 31942069), and the transcriptional activation of PD-L1 by NF- κ B in triple-negative breast cancer has been intensively studied (PMID: 29263152). However, as we previously mentioned in our results, changes in TMUB1 levels did not influence the transcriptional levels of PD-L1 in the triple-negative breast cancer cell line MDA-MB-231 (Response Figure. 2a and 2b). Further, we found that TMUB1 did not significantly regulate TRAF6 in MDA-MB-231 (Response Figure. 2c), and therefore we concluded that TMUB1 does not pass through TRAF6 in order to affect PD-L1 levels. The binding of TRAF6 to HUWE1 has been reported previously (PMID: 27746020) and we verified this binding (Response Figure. 2d), however, the binding of both results in the generation of a K48 branch by HUWE1 on the K63 chain formed by TRAF6, again involving NF- κ B activation (PMID: 27746020), and therefore although there is an association between the two, it does not involve the TMUB1-HUWE1 pathway in PD-L1 regulation that we have found.

Response Figure. 2

(a) The qRT-PCR analysis of the mRNA expression of PD-L1 in the MDA-MB-231 cells with stable Flag-tagged empty-vector (EV) or TMUB1-Flag expression. Data presented as mean \pm SEM. of n=3. Mann-Whitney test; n.s = no significance.

(b) The qRT-PCR analysis of the mRNA expression of PD-L1 in control or TMUB1-knockdown (sh-TMUB1#1 and sh-TMUB1#2) MDA-MB-231 cells. Data presented as mean \pm SEM. of n=3. One-way ANOVA followed by Tukey test; n.s = no significance.

(c) Immunoblots of TRAF6 in the MDA-MB-231 cells with stable Flag-tagged empty-vector (EV) or TMUB1-Flag expression.

(d) Co-IP analysis of the interaction between endogenous HUWE1 and endogenous TRAF6 within MDA-MB-231 cells. IgG was used as the negative control.

PD-L1 ubiquitination and degradation by E3 ligase/s have been reported previously. Authors may want to cite a recent paper showing that PD-L1 is ubiquitinated and degraded by the E3 ligase FBXO22. (De S, Holvey-Bates EG, Mahen K, Willard B, Stark GR. The ubiquitin E3 ligase FBXO22 degrades PD-L1 and sensitizes cancer cells to DNA damage. PNAS 2021, 23;118(47):e2112674118).

Response:

Thanks for your helpful suggestion! We have added the reference in the appropriate place in the manuscript (**Line 90**).

2. Mezzadra et al. (Identification of CMTM6 and CMTM4 as PD-L1 protein regulators. Nature, 549(7670), 106–110. 2017), ref 37, reported that the cellular membrane protein CMTM4/6 interacts with PD-L1, leading to inhibition of PD-L1 ubiquitination and degradation, which consequently impaired T cell activity. Is there any cross-talk between TMUB1 and CMTM6 or CMTM4 in the process of inhibition of PD-L1 degradation?

Response:

Thank you for your useful recommendation! We discovered that TMUB1 does not directly bind CTMT4 and CTMT6 (**Response Figures 3a and 3b**), nor does it affect the protein quantity or RNA abundance of CTMT4 and CTMT6 (**Response Figure. 3c-3e**). In contrast, neither CMTM4 nor CMTM6 influence the RNA or protein abundance of TMUB1 (**Response Figure. 3f-3i**). Considering that TMUB1 primarily performs its activity in the endoplasmic reticulum, CMTM4/6

impacts the destiny of the PD-L1 protein following its exit from the endoplasmic reticulum. We therefore believe that a direct contact between the two pathways is unlikely.

Response Figure. 3

- (a) Co-IP analysis of the interaction between endogenous TMUB1 and endogenous CMTM4 within MDA-MB-231 cells. IgG was used as the negative control.
- (b) Co-IP analysis of the interaction between endogenous TMUB1 and endogenous CMTM6 within MDA-MB-231 cells. IgG was used as the negative control.
- (c) Immunoblots of CMTM4 and CMTM6 in the MDA-MB-231 cells with stable Flag-tagged empty-vector (EV) or TMUB1-Flag expression.
- (d) The qRT-PCR analysis of the mRNA expression of CMTM4 in the MDA-MB-231 cells with overexpression of TMUB1 or vector. Mann-Whitney test, n=5, n.s = no significance.
- (e) The qRT-PCR analysis of the mRNA expression of CMTM6 in the MDA-MB-231 cells with overexpression of TMUB1 or vector. Mann-Whitney test, n=5, n.s = no significance.
- (f) Immunoblots of TMUB1 in the MDA-MB-231 cells with Flag-tagged empty-vector (EV) or CMTM4-Flag expression. Mann-Whitney test, n=5, n.s = no significance.
- (g) Immunoblots of TMUB1 in the MDA-MB-231 cells with Flag-tagged empty-vector (EV) or CMTM6-Flag expression. Mann-Whitney test, n=5, n.s = no significance.
- (h) The qRT-PCR analysis of the mRNA expression of TMUB1 in the MDA-MB-231 cells with overexpression of CMTM4 or vector. Mann-Whitney test, n=5, n.s = no significance.
- (i) The qRT-PCR analysis of the mRNA expression of TMUB1 in the MDA-MB-231 cells with overexpression of CMTM6 or vector. Mann-Whitney test, n=5, n.s = no significance.

3. The authors showed that TMUB1 could bind directly to the glycosylation mutant PD-L1 (4NQ) and reduce the degradation of mature PD-L1 as well as that of non-glycosylated PD-L1. They also showed that HUWEI binds to glycosylated and non-glycosylated forms of PD-L1 upon MG132

treatment. Does HUWE1 bind to 4NQ mutant PD-L1? Can over expression of HUWE1 in 4NQ mutant cells increase PDL1 degradation and antagonize TMUB1 mediated inhibition of the degradation of PD-L1?

Response:

Thanks for your helpful suggestion! HUWE1 directly binds to 4NQ mutant PD-L1 (**Response Figure. 4a**), overexpression of HUWE1 considerably lowers the abundance of 4NQ mutant PD-L1 (**Response Figure. 4b**), and overexpression of HUWE1 inhibits the elevation of 4NQ PD-L1 abundance mediated by TMUB1 (**Response Figure. 4c**). Consistent with our earlier hypothesis, this shows that HUWE1 is directly involved in the regulation of non-glycosylated PD-L1 breakdown.

Response Figure. 4

- (a) Co-IP analysis of the interaction between endogenous HUWE1 and PD-L1-4NQ-HA within MDA-MB-231 cells. IgG was used as the negative control.
- (b) Immunoblots of PD-L1-4NQ-HA in the MDA-MB-231 cells with Flag-tagged empty-vector (EV) or HUWE1-Flag expression.
- (c) Immunoblots of PD-L1-4NQ-HA in the MDA-MB-231 cells with Myc-tagged TMUB1 or Flag-tagged HUWE1 overexpression.

4. The peptide PTPR decreases PD-L1 expression. The authors may need to check whether PTPR increases the expression level of HUWE1 to facilitate the degradation of PD-L1.

Response:

Thanks for your helpful suggestion! PTPR treatment had no effect on the protein abundance of HUWE1 (**Response Figure 5a**) or the RNA abundance of HUWE1, PD-L1, and TMUB1 (**Response Figure 5b**), but it did increase the binding of HUWE1 to PD-L1 (**Response Figure. 5c**), which is consistent with our earlier idea and, coupled with our previous findings, suggests that PTPR should function by altering the interaction between TMUB1 and PD-L1 and boosting the binding of HUWE1.

Response Figure. 5

- (a) Immunoblots of HUWE1 in the MDA-MB-231 cells treated with 10 μ M of PTPR for 12 hr or not.
- (b) The qRT-PCR analysis of the mRNA expression of PD-L1/TMUB1/HUWE1 in the MDA-MB-231 cells treated with 10 μ M of PTPR for 12 hr or not. Mann-Whitney test, n=5, n.s = no significance.
- (c) Co-IP analysis of the interaction between endogenous HUWE1 and PD-L1 within MDA-MB-231 cells treated with 10 μ M of PTPR for 12 hr or not. IgG was used as the negative control.

5. Figure 3 a and b results were shown in HEK293T cells, not in MDAMB231 cells. Needs to be fixed in the legend.

Response:

Sorry for any confusion. We have made corrections in the revised manuscript.

In Summary, we thank the Reviewer for requesting several important and specific experiments that have permitted more rigorous, qualitative conclusions in support of the model proposed, and which will clearly enhance the confidence of the conclusions that can be drawn.

Reviewer #3 (expertise in HOPS/TMUB1, inflammation):

Please find enclosed the comments of the paper entitled: Break Antitumor Immunosuppression by Targeting TMUB1, a Novel Modulator for Posttranslational Modifications of PD-L1, by Shi C, – Aifu Lin et al., that has been submitted for publication in Nature Comm.

The research shows a novel target, PD-L1, of TMUB1 modifier in a context of tumor cells.

Indeed, the manuscript is difficult to read and understand. A huge series of experiments are present, but a lot of them are not useful and create confusion. The controls of a lot of experiments are not present and the quality of the WB and the images of IF are not well performed. To define a role of a protein is not possible to perform almost all experiments by transfection and analyze by tag the expression. These ways to perform experiments alter completely the biology and physiology of the cells.

The aim of the manuscript is loss in a lot of streams, but does not go straight to the question. The result of the ability of TMUB to control PD-L1 stability is present, but the mechanism is far to be explained. Also, the potential use of the peptide is not convincing the idea about the HUWE is not well determined.

Response:

In our study, we identified TMUB1, a potential regulatory protein of PD-L1, through immunoprecipitation tandem mass spectrometry coupled with clinical data analysis, and conducted a series of experiments to elucidate the specific mechanism and biological significance of TMUB1 regulation of PD-L1. The mechanism of competitive binding of TMUB1 to HUWE1 in the ubiquitinated degradation of PD-L1 was elucidated, as well as the protection of non-glycosylated immature PD-L1 by TMUB1 binding and promotion of STT3A with PD-L1 to enhance its glycosylation. The key role of TMUB1 in tumor immune evasion was also elucidated with the help of a series of animal models and clinical samples. Further, we designed a peptide PTPR based on the above mechanism, which inhibits tumor immune evasion by blocking the binding of PD-L1 to TMUB1 and promoting the HUWE1-mediated degradation of PD-L1 ubiquitination through a sequence similar to TMUB1.

We would also like to thank you for your valuable comments and we have made certain additions and modifications to the data we have presented based on your comments, including the addition of appropriate controls, more detailed experimental descriptions, further elucidation of the PTPR mechanism, and analysis of the binding and abundance of endogenous proteins. We believe that with your guidance, these new additions will make our study more convincing.

Technical Analysis

The technical analyses of the manuscript showed a high number of flaws, almost all experiments lack of controls and the majority of the experiments have been performed in transfection. Even the experiment to test the half-life of PD-L1 in presence of TMUB1 has been performed in transfection. This is not corrected way to establish protein stability.

Response:

Thank you for your valuable recommendation! Overexpression transfection and knockdown are the most typical methods for investigating the function of a protein, and are widely recognized in the research of PD-L1 regulatory proteins (**PMID: 29160310, 30397328, 30952982, 33879767**). In addition to avoiding the effects of transfection, all cell lines employed express the target gene in a stable manner, and our tests involve both overexpression and knockdown. Simultaneously, the outcomes of our animal tests and clinical samples have partially verified our cellular discoveries, ensuring the validity of the pathways we have found.

In almost all of the WB are not present Molecular Weight Markers in supplementary data.

Response:

Sorry for any confusion. We have marked all molecular weight markers in the revised manuscript.

Arbitrarily, they displayed different bands of TMUB1 and PD-L1 in relation to the experiments performed.

Response:

We apologize for any confusion. For TMUB1, we observed a band about 25 KD that is significantly diminished in knockdown cell lines compared to KO cell lines. As for PD-L1, since

the study process is progressive due to our understanding of the mechanism of TMUB1 regulation of PD-L1, as revealed in previous studies, it is the mature PD-L1 that primarily functions, with the location of the bands primarily around 50 KD. However, since we discovered that TMUB1 also has an important regulatory function for non-glycosylated PD-L1, we also studied the immature PD-L1 around 33 KD in Figure 3 To help illustrate this, we have included more detailed marker comments to all Western blot data and a more detailed discussion in the Results section to prevent misinterpretation; thank you for bringing this to our attention.

The survival test of T-cells has been defined, but the why the cells died is not described, apoptosis? No experiments are reported

Response:

Sorry for any confusion. We performed flow cytometry analysis and determined that overexpression of TMUB1 reduced apoptosis in tumor cells co-cultured with T cells (**Response Figure. 6a**), indicating that TMUB1 regulates PD-L1 stability and attenuates the killing of tumor cells by T cells.

Response Figure. 6

(a) Flow cytometric analysis of the PI⁺ and Annexin V⁺ cells in Sh-TMUB1 or Sh- Scramble MDA-MB-231 cell lines co cultured with activated T cells for 48h or not. The ratio of MDA-MB-231 cells to T cells: 1:3.

In some IF experiments assessing intracellular localization, the merged images of TMUB and PDL1 or HUWE1 are not reported (Fig3C, D and Ext data Fig2A). The merge with compartmental marker of each of the partner one by one are not enough to assess their interaction, they MUST be revealed in the same specimen.

Response:

Apologies for any confusion, and thank you for your helpful advice. We focused primarily on the effect of TMUB1 overexpression on membrane PD-L1 abundance for **Extended Data Figure.2A**. The IP versus pulldown experiment clearly demonstrates the binding of TMUB1 and PD-L1, including interactions of endogenous proteins, interactions of exogenously overexpressed proteins, and interactions of in vitro purified proteins (**Figure 1h and Extended Data Figure 1c and 1e**).

In addition, the distribution of PD-L1, HUWE1, and TMUB1 in the endoplasmic reticulum has been concurrently supported by several organelle isolation techniques. IP Assay for the isolation of

endoplasmic reticulum fractions also supports the binding of PD-L1 to HUWE1/TMUB1 on the endoplasmic reticulum (**Figure 3e and 3f and Extended Data Figure 4k and 4l**). Due to the limited availability of antibody species, the endogenous triple labeling of Calnexin+PD-L1+TMUB1/HUWE1 cannot be done.

The compartmental marker HAS NOT TO BE TRANSFECTED OR TAGGED for detection in IF (Fig3C, D). Why Calnexin-GFP has been used? Moreover, the GFP creates an ER stress of the cells.

Response:

Thanks for your helpful suggestion. We re-stained endogenous Calnexin with antibody to determine the endoplasmic reticulum localization of TMUB1 or HUWE1 (**Response Figure. 7a and 7b**).

When analysing the internalization of PTPR a plasma membrane marker is required to verify the real intracellular localization (Fig 6f).

Response:

Thanks for your helpful suggestion. Our previous experiments used proteinase K to eliminate extracellular PTPR, so all that could be stained should be PTPR that successfully entered the cell, and we stained the cell membrane with NF2 to re-verify the distribution of PTPR in the cell (**Response Figure. 8a**).

Even if when transfected the IF detection does not interest TMUB, a control assessing TMUB amount is mandatory (WB or qPCR as preferred) (I.e., Ext data Fig2, 3). This is required for all proteins modulated but not detected in IF assays.

Response:

Sorry for any confusion. The cell lines used in our immunofluorescence experiments were all cell lines stably expressing TMUB1 or TMUB1 knockdown, of which WB results have been shown in **Extended Data Fig. 1h and 1i**, and we also supplemented the TMUB1 RNA abundance detection by RT-qPCR in TMUB1- overexpression or sh-TMUB1 cell lines (**Response Figure. 9a and 9b**).

In some immunofluorescent assay scale bars are not reported or, if present, not in all the squares of the images (i.e., Fig 6f, Ext data Fig2A or Ext data Fig3C, D). They must be the same in all of them. Pay attention and choose one format only. I know that captions have to be short, but if too short they are no useful at all.

Response:

Sorry for any confusion and thanks for your helpful suggestion. We have made corrections in the revised manuscript.

Conceptual Analysis:

The data that TMUB acts on PDL1 stability seems certain, but the mechanistic explanation of

TMUB activity is quite confused and unclear. Many inhibitors or other factors used in the experimental procedure are only cited, sometimes without introducing their function or the reason for use.

Response:

Sorry for any confusion.

As a proteasome inhibitor, MG132 effectively inhibits the ubiquitous degradation of PD-L1, allowing us to detect the degree of PD-L1 ubiquitination under various conditions.

Chloroquine is an inhibitor of intracellular lysosomes and we used it to exclude the regulation of TMUB1 in the PD-L1 lysosomal degradation pathway.

PNGase F cleaves between N-acetylglucosamine (GlcNAc) and aspartyl residues in the innermost part of the high mannose, hetero- and complex oligosaccharide fractions of N-linked glycoproteins, enabling complete in vitro deglycosylation of PD-L1 to detect the total abundance of PD-L1.

Tunicamycin inhibits N-glycosylation and blocks GlcNAc phosphotransferase (GPT) intracellularly and was used to validate the function of TMUB1 in the degradation and glycosylation of non-glycosylated PD-L1.

Eeyarestatin I (Eer I) is a potent inhibitor of endoplasmic reticulum-associated protein degradation (ERAD). We used it to validate the regulatory role of TMUB1 in PD-L1 endoplasmic reticulum-associated protein degradation.

We have made additional notes in the **Results and Legends** sections in the revised manuscript.

“TMUB1 competes with HUWE1, a novel E3 ubiquitin ligase of PD-L1, to interact with PD-L1 and attenuates its polyubiquitination “

If assessing the competition between TMUB and HUWE in PDL1 site of binding, the effect must be observed in both TMUB vs HUWE and HUWE vs TMUB direction...in this paper only how TMUB hinders HUWE binding is observed, but not the reverse reaction.

Response:

Thanks for your helpful suggestion. We initially duplicated the tests with exogenous protein Co-IP assay with endogenous protein Co-IP assay, taking your earlier recommendations into

consideration (**Response Figure. 10a**). As recommended, we conducted tests and demonstrated that HUWE1 overexpression inhibited the binding of TMUB1 to PD-L1 (**Response Figure 10b**), suggesting a competitive binding interaction between HUWE1 and TMUB1 to PD-L1.

Response Figure. 10

- (a) Co-IP analysis of the interaction between endogenous HUWE1 and PD-L1 within MDA-MB-231 cells with TMUB1 overexpression or not. IgG was used as the negative control.
- (b) Co-IP analysis of the interaction between endogenous HUWE1 and PD-L1 within MDA-MB-231 cells with TMUB1 knockdown or not. IgG was used as the negative control.
- (c) Co-IP analysis of the interaction between endogenous TMUB1 and PD-L1 within MDA-MB-231 cells with HUWE1 overexpression or not. IgG was used as the negative control.

Moreover, TMUB1, as modifier, to be linked to PD-L1 needs an E3, and the authors hypothesize HUWE only such a sort of competitor. Indeed, the author cites a manuscript, where HOPS/TMUB1 regulates TRAF6, but in this case HOPS/TMUB1 does not act as a competitor, but such as a regulator of NF- κ B. No discussion or scenario are suggested by the author about a possible mechanism of the PD-L1-TMUB1 interaction as a modifier.

Response:

Sorry for any confusion and thanks for your helpful suggestion. Our research demonstrates the regulatory role of TMUB1 in the post-translational modification of PD-L1, which enhances PD-L1 protein abundance. TRAF6 activation in the NF- κ B signaling pathway has been extensively examined (PMID: 29894691, 31942069), as has the transcriptional activation of PD-L1 by NF- κ B in triple-negative breast cancer (PMID: 29263152). Changes in TMUB1 levels did not influence the transcriptional levels of PD-L1 in the triple-negative breast cancer cell line MDA-MB-231, as previously indicated in our data (Response Figure. 11a and 11b). Further, we discovered that

TMUB1 did not significantly influence TRAF6 in MDA-MB-231 (**Response Figure 11c**), which may be related to different cell types or tumor selectivity; consequently, we concluded that TMUB1 does not impact PD-L1 levels via the TRAF6 or NF- κ B pathway.

Obviously, we discovered that TMUB1 regulates PD-L1 activity in numerous cancer types, and for cancers in which TMUB1 activates NF-B, targeting TMUB1 has the ability to achieve dual inhibition from RNA and protein abundance, a mechanism that strengthens the therapeutic potential of TMUB1. In addition, we have included pertinent information to the Discussion section of the revised text.

Response Figure. 11

(a) The qRT-PCR analysis of the mRNA expression of PD-L1 in the MDA-MB-231 cells with stable Flag-tagged empty-vector (EV) or TMUB1-Flag expression. Data presented as mean \pm SEM. of n=3. Mann-Whitney test; n.s = no significance.

(b) The qRT-PCR analysis of the mRNA expression of PD-L1 in control or TMUB1-knockdown (sh-TMUB1#1 and sh-TMUB1#2) MDA-MB-231 cells. Data presented as mean \pm SEM. of n=3. One-way ANOVA followed by Tukey test; n.s = no significance.

(c) Immunoblots of TRAF6 in the MDA-MB-231 cells with stable Flag-tagged empty-vector (EV) or TMUB1-Flag expression.

“Mutation analysis revealed that K178 and K281 were associated with PD-L1 protein stability and ubiquitination, while only K281, but not K75, K178, K185, or K280 (Extended Data Fig. 3g), is required for the HUWE1-induced ubiquitination and degradation of PD-L1 (Extended Data Fig.3h-j)”

Several errors are present in this sentence. 1) Any K at 280 position is reported; 2) of the two lysines the K178 rather than the K281 is involved in degradation. If K281 was the effective one, its mutation should make TMUB ineffective and deplete its function... on the contrary PDL1 ubiquitination is still present in the absence of K281. Vice versa and for the same reason K178,

whose mutation deplete PDL1 degradation, is the functional one.

Response:

Sorry for any confusion. Thank you for pointing out the above issue, question 1 was an error in our writing process. Issue 2 is an error in the assembly and labelling of the images. We mislabelled the two groups, with lane 3/4 belonging to K178R and lane 5/6 belonging to K281R, which is in line with the predominant site being listed first (**Response Figure. 12a**). We have actually obtained similar results in our previous exogenous PD-L1 IP experiments (**Response Figure. 12b**), for which we apologize and have corrected them in the revised manuscript.

Response Figure. 12

(a) Co-IP analysis of the interaction between HA-UB and WT/K178R/K281R PD-L1-Flag in HEK-293T cells.

(b) TUBE-pull-down analysis of the interaction between PD-L1-Wild Type (WT) or K281R/K178R mutant-Flag and ubiquitin with or without HUWE1-overexpression in HEK-293T cells.

The TUBE assays used to perform ubiquitination analysis deserves quite more details.

Response:

Sorry for any confusion. We have supplemented the relevant experimental content in the Methods section (**Page 33**).

Only Fig3N gives a clear idea of an effect of HOPS in affecting the balance between glycosylated and not glycosylated forms of PDL1. The TMUB relationship with STT3A is assessed by the Co-IP binding evaluated, as nearly always in this paper, using transfected and tagged proteins and bioinformatics. In the paper isn't reported any experiment suggesting a clear interaction TMUB-STT3A in controlling PDL1 glycosylation (i.e., glycosylation rate STT3A-dependent in the lack or overexpression of TMUB).

Response:

Sorry for any confusion. To further discover the difference in PD-L1 glycosylation due to TMUB1, the binding of endogenous TMUB1 and PD-L1 to STT3A was demonstrated (**Response Figure. 13a, 13b**). The promotion of PD-L1 glycosylation by STT3A in tumors has been widely reported (**PMID: 31305264, 29765039**), and in line with this, we found that both PD-L1 and TMUB1 bind directly to STT3A, while overexpression of TMUB1 enhances the binding of PD-L1 to STT3A (**Response Figure. 13c**). We have also found that knockdown of TMUB1 inhibited the transition from non-glycosylation to glycosylation of PD-L1 and counted the ratio of the two to further quantify the difference in PD-L1 glycosylation due to TMUB1 (**Response Figure. 13d**), and therefore we suggest that TMUB1 promotes the glycosylation of PD-L1 by enhancing the interaction between STT3A and PD-L1.

Response Figure. 13

(a) Co-IP analysis of the interaction between endogenous PD-L1 and endogenous STT3A within MDA-MB-231 cells. IgG was used as the negative control.

(b) Co-IP analysis of the interaction between endogenous TMUB1 and endogenous STT3A within MDA-MB-231 cells. IgG was used as the negative control.

(c) Co-IP analysis of the interaction between endogenous PD-L1 and endogenous STT3A within MDA-MB-231 cells with TMUB1-Overexpression or not. IgG was used as the negative control.

(d) Immunoblots of PD-L1 in TMUB1-knockdown or control MDA-MB-231 cells following treatment with 10 μ M of MG132 and 50 μ M of CQ for 6 hr. Cell lysates were treated with PNGase F. Triangle: Glycosylated PD-L1, Circle: Non-glycosylated PD-L1.

Moreover (pag.17 lines 328), the authors wrote Together, TMUB1 regulated the glycosylation of PD-L1 by recruiting STT3A.... What means recruiting a direct binding a complex?

Response:

Sorry for any confusion. We have changed the statement to "TMUB1 enhances the binding of PD-L1 to STT3A" to more correctly reflect our results.

Pag. 18 The experiment in mice is not well described, the reader is misled by the description. The cell lines and the mice used are not well explicated.

Response:

Given the differences in mouse strains, we chose the EO771 mouse tumor cell line (syngeneic B16) with C57BL/6 mice for the same treatment and experiments as the 4T1 mouse breast cancer cell line and Balb/C mice. Similarly, we used the gastric cancer cell lines MFC in Balb/C mouse for tests to evaluate the involvement of Tmub1-targets in the control of tumor immunity in various cancer types. Targeting Tmub1 resulted in strong antitumor immunological responses in many cell lines and mice, indicating that TMUB1 is a viable target for cancer immunotherapy. In the revised manuscript, fixes and further clarifications have been made.

“the peptide capable of inhibiting the binding of TMUB1 to PD-L1 were designed”

The PTPR inhibiting peptide developed through the analysis of binding sites, is produced to counteracts the inhibiting effect of TMUB on HUWE mediate degradation. Even if in vivo xenografts experiments give encouraging results and suggest an antitumoral effect in mice, the mechanism is not clearly explained.

“Indeed, the addition of PTPR weakened the binding between PD-L1 and TMUB1 and reduced the level of PD-L1 protein caused by TMUB1 overexpression (Fig. 6h and Extended Data Fig. 9e)”

“PTPR was a TMUB1-targeting molecule that effectively inhibits the interaction between PD-L1 and TMUB1 and downregulates the cellular abundance of PD-L1 (Fig.6k).”

The authors wrote the opposite in the first part...

Response:

Sorry for any confusion. What we want to show in these two paragraphs is that PTPR reduces cellular PD-L1 abundance by inhibiting the binding of TMUB1 to PD-L1, thereby preventing

TMUB1 from properly regulating PD-L1. We have rearranged the formulation to avoid ambiguity and thank you for pointing it out.

The peptide binds to the shared PDL 1 binding site. If so, it would inhibit HUWE1 binding and further degradation thus enhancing half-life. Why are these data not included? On the contrary the author showed an enhanced ubiquitination rate upon PTPR addition in cell cultures. Why if binding site is inhibited by PTPR the author declares that PDL1 levels are reduced by increased degradation?

This question deserves deeper analysis and is not answered in the submitted paper.

Response:

Sorry for any confusion and thanks for your helpful suggestion. We performed molecular docking simulations (due to the excessive length of the HUWE1 sequence, we used its protein-binding domain 1603-1990 for substitution, **PMID: 34314700**). The results show that TMUB1/PTPR does not bind the exact same PD-L1 sequence as HUWE1, and also that the binding interface is significantly different (**Response Figure.14a**). Also the disparity in molecular size between the three (PTPR 15AA vs. TMUB1 246AA vs. HUWE1 4374AA) makes it also feasible to take into account the disparate molecular size difference between the three, making it possible for PTPR to affect the binding of TMUB1 to PD-L1, without blocking the binding of PD-L1 to HUWE1. Further, we verified this and PTPR treatment did enhance the binding of HUWE1 to PD-L1 (**Response Figure.14b**), which is also consistent with our previous finding of enhanced ubiquitination and degradation of PD-L1 due to PTPR (**Response Figure.14c and 14d**). Furthermore, PTPR treatment did not affect HUWE1/TMUB1/PD-L1 RNA levels and HUWE1 protein abundance (**Response Figure.14e and 14f**), but significantly attenuated intracellular PD-L1 abundance (**Response Figure.14g**). Based on the above, we summarize the working model of PTPR as follows. PTPR weakens the binding of TMUB1 to PD-L1 through a PD-L1 binding domain similar to TMUB1, but does not block the binding of PD-L1 to HUWE1 due to the small spatial site resistance and different interactions interface, thus promoting the interactions between PD-L1 and HUWE1, leading to the ubiquitinated degradation of PD-L1 (**Response Figure.14h**). Of course, Subsequent access to the crystal structure of any two complexes or three might adequately address

this issue, but there is a real lack of better methods for testing at this stage, as we explain in the **Discussion** section.

Response Figure. 14

(a) HUWE1-WWE-PD-L1 (upper), TMUB1-PD-L1 (middle) and PTPR-PD-L1 (lower) complexes obtained on the basis of molecular docking. The yellow dashed line shows hydrogen bonding and the burgundy dashed line shows salt bridging.

(b) Co-IP analysis of the interaction between endogenous HUWE1 and PD-L1 within MDA-MB-231 cells treated with 10 μ M of PTPR for 12 hr or not. IgG was used as the negative control.

(c) Co-IP analysis of the interaction between endogenous ubiquitin and PD-L1 within MDA-MB-231 cells treated with 10 μ M of PTPR for 12 hr or not. IgG was used as the negative control.

(d) Immunoblots of PD-L1 in MDA-MB-231 cells treated with 10 μ M of PTPR following treatment with 20 μ g/mL cycloheximide (CHX) for the indicated time points.

(e) Immunoblots of PD-L1 in the MDA-MB-231 cells treated with 10 μ M of PTPR for 12 hr or not.

(f) The qRT-PCR analysis of the mRNA expression of PD-L1/TMUB1/HUWE1 in the MDA-MB-231 cells treated with 10 μ M of PTPR for 12 hr or not. Mann-Whitney test, n=5, n.s = no significance.

(g) Immunoblots of HUWE1 in the MDA-MB-231 cells treated with 10 μ M of PTPR for 12 hr or not.

(h) Schematic diagram for the PTPR peptide.

In conclusion, we appreciate the Reviewer's constructive comments on the experiments, which will result in a more robust and accessible manuscript, as well as the request for several crucial and specific experiments that have allowed for more rigorous, qualitative conclusions in support of the proposed model, and which will unquestionably increase the confidence in the conclusions that can be drawn.

REVIEWERS' COMMENTS

Reviewer #1 (Remarks to the Author):

The authors have made a highly comprehensive, highly detailed response to prior critiques. My critiques and queries were addressed satisfactorily, or an adequate explanation for not being fully addressed was adduced. It appears that responses to other critiques were similarly thorough.

One minor point, consider changing "Break" to "Breaking: in the title

Reviewer #2 (Remarks to the Author):

The manuscript resubmitted by Chengyu Shi et al. has been thoroughly revised and has provided new data based on the suggestions of the previous review.

A few concerns remain:

1. Some of the new figures in the revised manuscript need to be numbered correctly. For example, extended data Fig. 4e, f in the revised manuscript is not in accordance with the numbering in the results.
2. The authors included new figures in the "Response to reviewers". However, they did not mention where those figures were incorporated in the revised manuscript. Therefore, it was troublesome to locate those figures in the manuscript because of lack of clarity in the response.

Reviewer #3 (Remarks to the Author):

Dear Authors,
please find enclosed the renewed comments to the manuscript: Break Antitumor Immunosuppression by Targeting TMUB1, a Novel Modulator for Posttranslational Modifications of PD-L1, by Shi C, -Aifu Lin et al.. I appreciated that the Authors replied to my comments and increased the quality of some experiments. Indeed, the previous experiments have been performed fast and inaccurate and now they have partial/mostly replied to my requests. I understand that the field is hot and the authors have to capitalize the important observation, but the biological mechanism by which TMUB1 regulates the stability of PD-L1 is far to be studied and explained.

Reviewer #1 (Remarks to the Author):

The authors have made a highly comprehensive, highly detailed response to prior critiques. My critiques and queries were addressed satisfactorily, or an adequate explanation for not being fully addressed was adduced. It appears that responses to other critiques were similarly thorough.

One minor point, consider changing "Break" to "Breaking: in the title

Response:

Thank you!

With the help of the editors, we have already changed our title to “Promoting anti-tumor immunity by targeting TMUB1, a modulator of PD-L1 polyubiquitination and glycosylation”.

Reviewer #2 (Remarks to the Author):

The manuscript resubmitted by Chengyu Shi et al. has been thoroughly revised and has provided new data based on the suggestions of the previous review.

A few concerns remain:

1. Some of the new figures in the revised manuscript need to be numbered correctly. For example, extended data Fig. 4e, f in the revised manuscript is not in accordance with the numbering in the results.

Response:

Sorry for our negligence. We have made corrections in the revised manuscript to ensure consistency between the Results and the Figures.

2. The authors included new figures in the “Response to reviewers”. However, they did not mention where those figures were incorporated in the revised manuscript. Therefore, it was troublesome to locate those figures in the manuscript because of lack of clarity in the response.

Response:

We apologized for the difficulties you may have experienced during the review process due to an oversight on our part.

Some of the results were not integrated into the final revised manuscript because they did not exactly match our content and because of the limitations of the length of the article.

The association of PD-L1 regulation by TMUB1 with CMTM4/CMTM6 is presented by **Supplementary Figure 2i-q**.

Regulation of PD-L1 4NQ mutants by HUWE1 is presented in **Supplementary Figure 4g, 4h and 4j**.

Evidence that PTPR does not regulate HUWE1 is presented in **Supplementary Figure 6e, f**.

Reviewer #3 (Remarks to the Author):

Dear Authors,

please find enclosed the renewed comments to the manuscript: Break Antitumor

Immunosuppression by Targeting TMUB1, a Novel Modulator for Posttranslational Modifications of PD-L1, by Shi C, –Aifu Lin et al.. I appreciated that the Authors replied to my comments and increased the quality of some experiments. Indeed, the previous experiments have been performed fast and inaccurate and now they have partial/mostly replied to my requests.

I understand that the field is hot and the authors have to capitalize the important observation, but the biological mechanism by which TMUB1 regulates the stability of PD-L1 is far to be studied and explained.

Response:

Thank you!

We will continue to explore this in depth in the hope that our research will eventually lead to clinical application.